# KDM2B regulates hippocampal morphogenesis by transcriptionally silencing Wnt signaling in neural progenitors

Bo Zhang[1], Chen Zhao[1], Wenchen Shen[1], Wei Li[1], Yue Zheng[1], Xiangfei Kong[1], Junbao Wang[1], Xudong Wu [2,3], Tao Zeng [4] ✉, Ying Liu [1] ✉ & Yan Zhou [1] ✉

The hippocampus plays major roles in learning and memory, and its formation requires precise coordination of patterning, cell proliferation, differentiation, and migration. Here we removed the chromatin-association capability of KDM2B in the progenitors of developing dorsal telencephalon (*Kdm2b^ΔCxxC*) to discover that *Kdm2b^ΔCxxC* hippocampus, particularly the dentate gyrus, became drastically smaller with disorganized cellular components and structure. *Kdm2b^ΔCxxC* mice display prominent defects in spatial memory, motor learning and fear conditioning, resembling patients with *KDM2B* mutations. The migration and differentiation of neural progenitor cells is greatly impeded in the developing *Kdm2b^ΔCxxC* hippocampus. Mechanism studies reveal that Wnt signaling genes in developing *Kdm2b^ΔCxxC* hippocampi are de-repressed due to reduced enrichment of repressive histone marks by polycomb repressive complexes. Activating the Wnt signaling disturbs hippocampal neurogenesis, recapitulating the effect of KDM2B loss. Together, we unveil a previously unappreciated gene repressive program mediated by KDM2B that controls progressive fate specifications and cell migration, hence morphogenesis of the hippocampus.

The hippocampus of the mammalian brain includes three major compartments: the hippocampus proper, which can be divided into three pyramidal subregions [cornu ammonis (CA) fields], the dentate gyrus (DG), and the subiculum. The hippocampus plays essential roles in spatial memory that enables navigation, in the formation of new memories, as well as in regulating mood and emotions[1]. Constructing a functional hippocampus requires precise production, migration, and assembly of a variety of distinct cell types during embryonic and early postnatal stages. Largely resembling neurogenesis of neocortical pyramidal neurons, the production of pyramidal neurons in CAs and granule cells in dentate gyrus follows the path of indirect neurogenesis[2], i.e., PAX6-expressing radial glial progenitor cells (RGCs) first give rise to TBR2[+] intermediate progenitor cells (IPCs),

which then produce NeuN[+] neurons. Particularly, the formation of the dentate gyrus relies on sequential emergence of germinative foci at different locations, including the dentate notch around embryonic (E) day 13.5 in mice, the fimbriodentate junction (FDJ) around E15.5 and the hilus around birth, which generate granule cells at different parts of DG[2,3]. Cell fate defects and skewed cell migration during hippocampal development underly a cohort of human neurologic and psychiatric diseases[4]. Moreover, the subgranule zone (SGZ) of adult DG contains neural stem cells (NSCs) to support neurogenesis, which might have implications in the formation of new memory.

The cell fate specification during development requires coordinated actions of transcription factors (TFs), epigenetic factors and *cis*-acting elements to ensure precise gene expression and silencing.

[1]Department of Neurosurgery, Medical Research Institute, Frontier Science Center of Immunology and Metabolism, Zhongnan Hospital of Wuhan University, Wuhan University, Wuhan, China. [2]Department of Cell Biology, Tianjin Medical University, Tianjin, China. [3]Department of Neurosurgery, Tianjin Medical University General Hospital, Tianjin, China. [4]Department of Neurosurgery, Shanghai Tenth People's Hospital, Tongji University School of Medicine, Shanghai 200072, China. ✉e-mail: ztbox73@163.com; y.liu@whu.edu.cn; yan.zhou@whu.edu.cn

Notably, repressive histone modifications including mono-ubiquitinated histone H2A at lysine 119 (H2AK119ub1) and trimethylated histone H3 at lysine 27 (H3K27me3), respectively modified by the Polycomb repressive complex 1 (PRC1) and PRC2, are essential fate regulators in embryogenesis, organogenesis and tissue homeostasis[5–9]. Ring1B, the ubiquitin ligase protein, is the core enzymatic component of all PRC1 complexes and plays multiple roles during neural development. The PRC1 can be recruited to the chromatin *via* either one of the chromobox proteins (canonical PRC1), or other adapter proteins including KDM2B (variant PRC1)[10–14]. Interestingly, PRC1 and PRC2 can reciprocally recognize repressive histone modifications mediated by each other or itself to stabilize repressive chromatin environments[15–19]. It has been reported that the PRC2 is required for hippocampal development and the maintenance of the adult neural stem cell pool of the DG, but the underlying cellular processes and molecular mechanisms were large elusive[20,21]. Furthermore, very little is known to what extent and how PRC1 is involved in hippocampal development.

KDM2B, previously known as JHDM1B, FBXL10, or NDY1, can recruit PRC1 to non-methylated CpG islands (CGIs), particularly at promoter regions, *via* its CxxC Zinc finger (ZF)[10–12,22,23]. The long isoform of KDM2B – KDM2BLF – contains a JmjC demethylation domain which removes the di-methylated lysine 36 of histone H3 (H3K36me2) to regulate pluripotency and early embryogenesis[24]. Mutations of human *KDM2B* gene are associated with neurodevelopment defects including intellectual disability (ID), speech delay, and behavioral abnormalities[25–27]. A recent study indicated that heterozygosity of *Kdm2b* in mice impaired neural stem cell self-renewal and leads to ASD/ID-like behaviors[28]. However, there is no in-depth analysis to dissect how PRC1- and/or demethylase-dependent roles of KDM2B participate in multiple facets of neural development, including self-renewal, migration, differentiation and localization of neural progenitors and their progeny.

We previously showed that KDM2B controls neocortical neuronal differentiation and the transcription of *Kdm2blf* is *cis*-regulated by a long non-coding RNA which is divergently transcribed from the promoter of *Kdm2blf*[29]. However, questions remain regarding to what extent and how KDM2B regulates neural development. Here we ablated the chromatin association capability of KDM2B in the developing dorsal forebrain to surprisingly find the morphogenesis of hippocampus, especially the DG, was greatly hampered. Moreover, intermediate progenitors could not properly migrate and differentiate upon dissociating KDM2B from the chromatin. The canonical Wnt signaling were aberrantly activated in mutant hippocampi, probably due to decreased enrichment of H2AK119ub and H3K27me3 in CGI promoters of key Wnt pathway genes.

## Results

### Removing KDM2B-CxxC causes hippocampal hypoplasia

KDM2B has two main isoforms: the long isoform KDM2BLF contains a demethylase JmjC domain while both isoforms share the CxxC zinc finger (ZF), the PHD domain, a F-box and the LRR domain. We generated the conditional knockout allele of *Kdm2b* by flanking exon 13 that encodes the CxxC ZF (75 amino acids, 8.3 kilodalton) with two *loxP* sequences (Supplementary Fig. 1a). These floxed *Kdm2b* mice, *Kdm2b^flox(CxxC)*, were crossed with *Emx1*-Cre and *Nestin*-Cre mice to generate conditional *Kdm2b^{Emx1-ΔCxxC}* and *Kdm2b^{Nestin-ΔCxxC}* conditional knockout (cKO) mice respectively to abolish KDM2B's association with the chromatin. Although *Kdm2b^{Nestin-ΔCxxC}* mice could not survive past postnatal day 7 (P7), *Kdm2b^{Emx1-ΔCxxC}* mice were born at the mendelian ratio and thrive through adulthood without gross abnormality (Supplementary Fig. 1b, c). RNA-seq of hippocampal neurospheres confirmed the deletion of exon 13 in cKO brains (Supplementary Fig. 1d). We then performed in situ hybridization of neonatal (P0) brains using probes targeting exon 13 (Supplementary Fig. 1e), which showed *Kdm2b* is expressed in the developing hippocampus across the CA

region and the DG of control *Kdm2b^{flox/flox}* brains (Supplementary Fig. 1f). *Kdm2b* is expressed at a higher level in the HP between the embryonic day (E) 16 and the P0 period (Supplementary Fig. 1i) and is present in IPCs of the developing DG (Supplementary Fig. 1j). Importantly, the expression of *Kdm2b-CxxC* was almost gone in the *Kdm2b^{Emx1-ΔCxxC}* hippocampi (Supplementary Fig. 1g), but is still present in regions where the *Emx1*-Cre is not active, such as the thalamus (Supplementary Fig. 1h). Moreover, immunoblotting of E15.5 *Kdm2b^{Nestin-ΔCxxC}* and P0 *Kdm2b^{Emx1-ΔCxxC}* neocortex revealed truncated long and short isoforms of KDM2B (Supplementary Fig. 1k–l), confirming the selective deletion of the CxxC ZF, which could abolish the CGI association by KDM2B and variant PRC1.1.

Strikingly, the hippocampi, particularly the DGs, of adult *Kdm2b^{Emx1-ΔCxxC}* cKO brains were greatly shrunk in size in all examined sections (Fig. 1a, Supplementary Fig. 2a, b). Although Wfs1-expressing pyramidal cells could be detected in the CA1 region of *Kdm2b^{Emx1-ΔCxxC}* cKO brains (Supplementary Fig. 2c), Calbindin-expressing cells, which regulates synaptic plasticity and memory[30], were mostly diminished (Fig. 1b). Both upper and lower blades of cKO DGs were much shorter than controls (Fig. 1c; Supplementary Fig. 2d–g), with numbers of NeuN+ granule cells and GFAP + SOX2+ NSCs decreased by 39.7% and 82.1% respectively (Fig. 1d–g). Immuno-staining showed irregularity and ectopic dispersion of Calbindin+, ZBTB20+ or NeuN+ granule cells in cKO DGs (Fig. 1b–d). ZBTB20 is expressed in immature projection neurons of the hippocampus and plays a critical role in hippocampal neurogenesis[31]. Fewer HopX+ NSCs, fewer TBR2+ neuroblasts, and fewer PROX1+ or DCX+ neurons were present in cKO SGZs (Fig. 1h; Supplementary Fig. 2h–2k). Cell densities of granule cells, newborn neurons, and NSCs were also greatly decreased in cKO DGs, partly reflecting the loosened cellular architecture on loss of KDM2B-CxxC (Supplementary Fig. 2l–2p). Interestingly, many HopX+ NSCs were found to be ectopically localized inside the granule cell layer of cKO DGs (Supplementary Fig. 2h, red arrows). The ventricles were also enlarged in cKO brains along with thinner neocortices (Fig. 1a, Supplementary Fig. 3a–c). The enlarged ventricles of cKO brains could already be seen at P0 for unknown reasons without thinning of neocortices (Supplementary Fig. 3d–g). No increase of apoptotic cells was detected in P7 cKO brains and hippocampi (Supplementary Fig. 3h, i). At P7, numbers of upper-layer (SATB2+) and lower-layer (CTIP2+) neocortical neurons were not significantly altered upon CxxC deletion of KDM2B (Supplementary Fig. 3j, k). Together, ablation of the chromatin association capability of KDM2B in developing dorsal forebrains causes hippocampal hypoplasia, while the thinning of adult cKO neocortices might be secondary to ventricle dilation.

### *Kdm2b-ΔCxxC* mice display defects in memory and learning

Since hippocampus is essential for spatial navigation and memory consolidation, as well as for exploration, anxiety, and depression behaviors[32–35], we conducted a series of behavior tests. First, Morris water maze tests revealed that *Kdm2b^{Emx1-ΔCxxC}* cKO mice were defective in spatial learning. It took cKO mice longer time to find the platform in the training stage (Fig. 2a, b). Consistently, in the probe trial on day 6, although control and *Kdm2b^{Emx1-ΔCxxC}* cKO mice showed no differences in swimming distances, velocity, and platform crossing times (Fig. 2c–f), cKO mice spent shorter time in quadrant holding the platform (Fig. 2g). Secondly, in contrast to control mice, *Kdm2b^{Emx1-ΔCxxC}* cKO mice failed to display prolonged freezing time in both context- and sound-induced fear conditioning tests (Fig. 2h–k). Third, the rotarod performance tests indicated that *Kdm2b^{Emx1-ΔCxxC}* cKO mice have compromised capacity for motor coordination and learning, as they endured shorter time in rotarods than controls (Fig. 2l). Interestingly, open field tests showed reluctance of *Kdm2b^{Emx1-ΔCxxC}* mice to explore center regions of open fields (Fig. 2m, n), which could be decreased willingness of cKO mice to explore and/or increased anxiety. The distance and velocity traveled, and immobility time in open

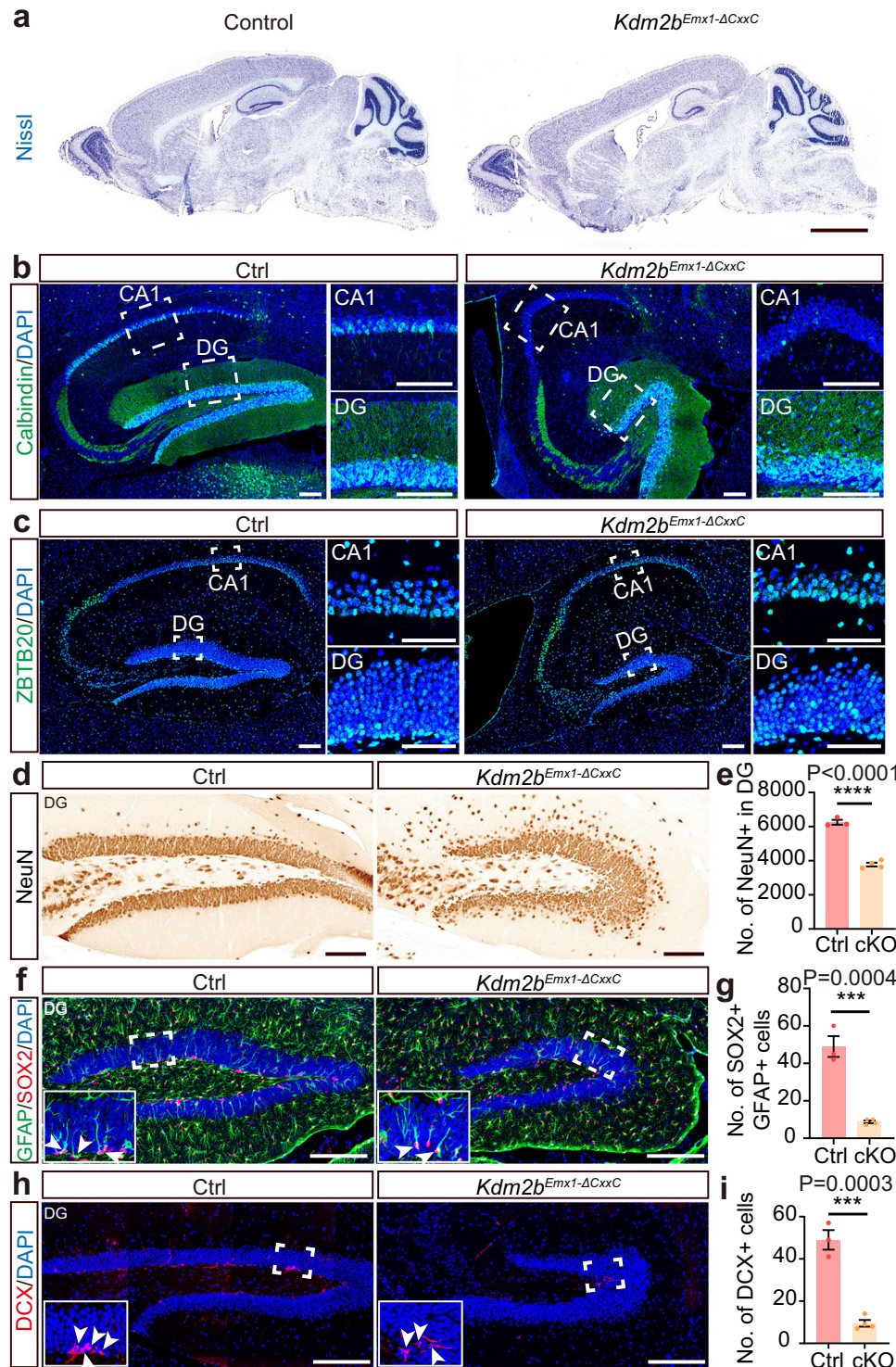

**Fig. 1 | Deletion of the KDM2B-CxxC causes hippocampal hypoplasia.**
**a** Representative images showing Nissl staining on sagittal sections of adult control and *Kdm2b^{Emx1-ΔCxxC}* brains. **b**, **c** Immunofluorescent (IF) staining of Calbindin (**b**) and ZBTB20 (**c**) on sagittal sections of adult control (left) and *Kdm2b^{Emx1-ΔCxxC}* (right) hippocampi. Nuclei were labeled with DAPI (blue). Boxed CA1 and dentate gyri (DG) were enlarged on the right. **d** Immunohistochemical (IHC) staining of NeuN on sagittal sections of adult control (left) and *Kdm2b^{Emx1-ΔCxxC}* (right) DG. **f**, **h** Double immunofluorescence of GFAP (green) and SOX2 (red) (**f**) and single immuno-fluorescence of DCX (**h**) on sagittal sections of adult control (left) and *Kdm2b^{Emx1-}*

*^{ΔCxxC}* (right) DG. Nuclei were labeled with DAPI (blue). Boxed regions were enlarged on bottom-left corners. Arrows denote GFAP + SOX2+ or DCX+ signals in the sub-granule zone (SGZ). **e**, **g**, **i** Quantification of NeuN+ cells in the DG (**e**), GFAP + SOX2+ cells in the SGZ (**g**) and DCX+ cells in the SGZ (**i**). *n* = 3 for control brains and *n* = 4 for *Kdm2b^{Emx1-ΔCxxC}* brains. Similar results were obtained for 3 control brains and 4 *Kdm2b^{Emx1-ΔCxxC}* brains (**a**–**c**). Data are represented as means ± SEM. Statistical significance was determined using an unpaired two-tailed Student's *t*-test (**e**, **g**, **i**). ***P < 0.001; ****P < 0.0001. Scale bars, 2 mm (**a**), 100 μm (**b**–**d**), 200 μm (**f**, **h**), 50 μm (**c** CA1 and DG).

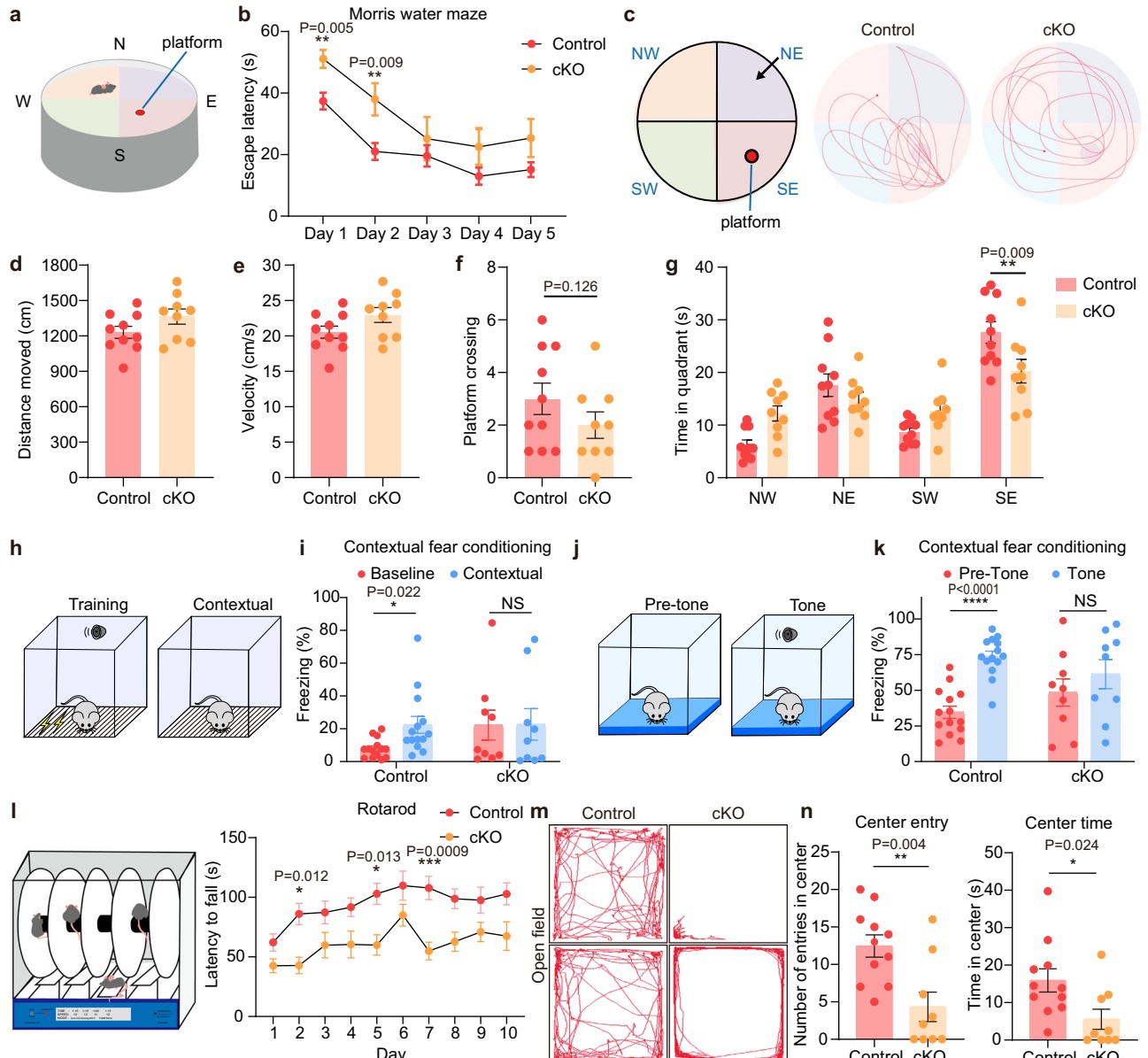

**Fig. 2 | *Kdm2b^(Emx1-ΔCxxC)* mice exhibit defects in spatial memory, contextual fear conditioning, and motor learning. a** Diagram of the Morris water maze test. **b** Latency to find the hidden platform across training period in the Morris water maze test. **c** An overhead view of the Morris water maze, and representative swim paths of control mice and *Kdm2b^(Emx1-ΔCxxC)* mice during the probe trial. The platform was set in the SE quadrant. **d**, **e** Distance moved (**d**) and velocity (**e**) during the probe trial (platform removed). **f** Frequencies of platform crossing during the probe trial. **g** Time spent in each quadrant during the probe trial. **h**, **i** The proportion of freezing time in context before training (Baseline) and after training

(Contextual). **j**, **k** The proportion of freezing time in a new context before tone (Pre-Tone) and after tone (Tone). **l** Latency to fall during the rotarod test. **m** Representative traces of control and *Kdm2b^(Emx1-ΔCxxC)* mice in the open-field arena. **n** Quantification of number of entries in center, and time spent in the center in the open-field test. Data are represented as means ± SEM. Statistical significance was determined using two-way ANOVA followed by Sidak's multiple comparisons test (**b**, **g**, **i**, **k**, **l**), or using an unpaired two-tailed Student's *t*-test (**d**–**f**, **n**). *$P < 0.05$, **$P < 0.01$, ***$P < 0.001$, and ****$P < 0.0001$; NS, not significant. $n = 10$ in (**a**–**g**), $n = 14$ in (**h**–**k**), $n = 11$ mice in (**l**–**n**) for control and $n = 9$ mice for *Kdm2b^(Emx1-ΔCxxC)*.

field tests were not significantly altered in *Kdm2b^(Emx1-ΔCxxC)* mice (Supplementary Fig. 4a). The immobility time of *Kdm2b^(Emx1-ΔCxxC)* mice in forced swimming tests were slightly shorter than control mice (no statistical significance), whereas the immobility time of cKO mice in tail suspension test was the same as controls, suggesting cKO mice did not display depression-related behavior (Supplementary Fig. 4b, c). Interestingly, cKO mice spent shorter time in open arms of elevated plus maze (no statistical significance), indicating that cKO mice tend to be hypersensitive to anxiety (Supplementary Fig. 4d). Together, ablation of the chromatin association capability of KDM2B leads to the developmental failure of the hippocampus, which might cause defects in spatial memory, fear conditioning, and motor learning.

### KDM2B-ΔCxxC has no effect on adult neurogenesis of the DG

The shrunken hippocampi and diminished NSC pool in *Kdm2b^(Emx1-ΔCxxC)* cKO mice prompted us to investigate whether the ablation of KDM2B's chromatin association capability could hamper adult neurogenesis of the DG. We thus crossed the *Kdm2b^(flox(CxxC))* mice with *Nestin-CreERT2* mice to produce *Kdm2b^(Nestin-CreERT2-ΔCxxC)* cKO mice. Reporter analyses indicated that *Nestin-CreERT2* is active in SGZ upon tamoxifen (TAM) induction (Supplementary Fig. 4e). Adult *Kdm2b^(Nestin-CreERT2-ΔCxxC)* cKO and control mice were administered with TAM for six consecutive days to ablate the CxxC ZF in adult NSCs and their progeny. BrdU was also administered for six consecutive days to label NSCs' progeny. Brains were collected at day 8 and day 35 of post-TAM injection

(Supplementary Fig. 4f, h) for immunofluorescent staining of BrdU, along with DCX (day 8) or PROX1 (day 35), two markers for new-born and mature granule neurons respectively. Data showed numbers of BrdU-labeled cells and DCX+BrdU+ double-positive cells, and ratios for DCX+BrdU+/BrdU+ cell were comparable between $Kdm2b^{Nestin-CreERT2-\Delta CxxC}$ cKO and control DGs in all examined sections at day 8 (Supplementary Fig. 4f–g, j–l). Similarly, at day 35, numbers of BrdU-labeled cells and PROX1+BrdU+ double-positive cells, and ratios for PROX1+BrdU+/BrdU+ cell were not altered in $Kdm2b^{Nestin-CreERT2-\Delta CxxC}$ cKO DGs (Supplementary Fig. 4h–i, m–o). Thus, removal of KDM2B's chromatin association capability in adult NSCs exerts no effect on adult neurogenesis of DGs. This could be due to the transient expression of KDM2B in embryonic and postnatal hippocampi. We would like to point out that the possibility of inefficient deletion of $Kdm2b$-CxxC on TAM treatment might exist.

## Hampered neurogenesis and migration upon KDM2B-ΔCxxC

To ask how DG neurogenesis was affected on loss of KDM2B-ΔCxxC, we first carried out EdU birthdating experiments by administering EdU at E15.5 followed by EdU and PROX1 co-labeling at P2. 31.3% fewer PROX1-labeled granule cells could be detected in cKO DGs but 259.0% more in the cKO FDJ. In cKO brains, 33.3% fewer E15.5-labeled EdU cells reached the DG and co-stained with PROX1, while ectopic PROX1 cells were detected at FDJ (Fig. 3a–c). Moreover, total number of PROX1+EdU+ cells was reduced by 27.1% in cKO hippocampi (Fig. 3d), with 39.5% fewer EdU+ cells in cKO DGs (Fig. 3e), indicating hampered neuronal production on loss of KDM2B-CxxC. Furthermore, proportions of PROX1+ cells in lower and upper blades of DGs were switched into a lower-more and upper-fewer status in cKO DGs (Fig. 3f–h). Congruently, by P7, a big chunk of PROX1+ cells could be seen at the FDJ of $Kdm2b^{Emx1-\Delta CxxC}$ brains (Supplementary Fig. 5a).

In DG development, neural progenitors, especially TBR2-expressing intermediate progenitor cells (IPCs) migrate from the dentate neuroepithelial (DNe) stem zone (the primary − 1ry matrix) through the dentate migratory stream (DMS, the secondary − 2ry matrix) to the forming DG (the 3ry matrix), while being distributed in multiple transient niches (Fig. 3l). The prominent DG defects in $Kdm2b^{Emx1-\Delta CxxC}$ cKO brains prompted us to examine distributions of intermediate progenitors and neurons along the migratory path. P0 brain sections were co-stained with TBR2 to label IPCs and with GFAP to label astrocytic scaffold at the fimbriodentate junction (FDJ) of the DMS and the fimbria. The total number of IPCs were increased by 15.3% upon the loss of KDM2B-CxxC. Strikingly, in $Kdm2b^{Emx1-\Delta CxxC}$ cKO brains, significant more TBR2+ IPCs were accumulated at the DNe and the DMS, but significant fewer TBR2+ IPCs were detected at the DG (Fig. 3i–m). Of note, TBR2+ IPCs were more loosely distributed at the FDJ, with the area of fimbria significantly decreased (Fig. 3i–k, n). Consequently, GFAP-labeled astrocytic scaffold scattered at the FDJ but constricted at the fimbria of $Kdm2b^{Emx1-\Delta CxxC}$ cKO brains (Fig. 3o), probably due to the compression by amassed TBR2+ cells.

The hampered migration of IPCs upon loss of KDM2B-CxxC could be due to defects of the cortical hem (CH) derived astrocytic scaffolds at the fimbria[36]. To exclude the possibility, $Kdm2b^{Nestin-\Delta CxxC}$ cKO brains were inspected, because reporter analyses validated that $Nestin$ is mostly not expressed in CH-derived astrocytic scaffolds, which are labeled by BLBP (Supplementary Fig. 5b). Data showed that P0 $Kdm2b^{Nestin-\Delta CxxC}$ brains displayed almost the same phenotypes as those in $Kdm2b^{Emx1-\Delta CxxC}$ brains, i.e., overproduction of TBR2+ IPCs, significantly more IPCs accumulated at the DNe and the FDJ, but fewer IPCs at the DG, suggesting that the migrating defects of IPCs upon loss of KDM2B-CxxC were not due to defects of CH-derived astrocytic scaffolds (Supplementary Fig. 5c–g). Together, loss of KDM2B-CxxC impedes migration of IPCs, subsequently resulting in hampered production and localization of granule neurons during hippocampal formation.

## KDM2B-ΔCxxC disturbs differentiation of neural progenitors

We next investigated how the RGC-IPC-Neuron neurogenesis path of hippocampi were influenced on loss of KDM2B-CxxC. At E13.5, when the hippocampal primordia just emerged, the distribution of BLBP, the marker for astrocytic progenitors at CH, were unaltered in $Kdm2b^{Emx1-\Delta CxxC}$ cKO brains (Supplementary Fig. 6a, b). Similarly, the expression pattern of SOX2, the marker for neocortical and hippocampal RGCs, were not changed in CH, DNe and hippocampal neuroepithelium (HNe) upon loss of KDM2B-CxxC (Supplementary Fig. 6a, the middle panel). In addition, the distribution of PAX6 and TBR2, markers for RGC and IPCs respectively, at E13.5 were almost the same between cKOs and controls (Supplementary Fig. 6a, the bottom panel). We further examined whether the capacity of proliferation and differentiation of hippocampal progenitors was affected at E14.5 by co-labeling brain sections with PAX6, TBR2 and EdU (2 h pulse) (Supplementary Fig. 6c). Data showed that deletion of KDM2B-CxxC had no effect on abundance and proliferation of PAX6+ RGCs and TBR2+ IPCs in DNEs (Supplementary Fig. 6d), HNE (Supplementary Fig. 6e). Consistently, the differentiation capability from RGCs to IPCs was unchanged in cKO brains, because the number of PAX6 + TBR2+ cells and the ratio of PAX6 + TBR2+ cells among all PAX6+ cells were comparable between cKOs and controls (Supplementary Fig. 6c–e). Thus, the hippocampal hypoplasia in $Kdm2b^{Emx1-\Delta CxxC}$ cKO brains was not due to specification, maintenance, and differentiation of early neural progenitors.

We then moved onto E16.5, when migration and neurogenesis of hippocampal progenitors are prominent. EdU was administered 2 h before sacrifice to label dividing cells. In control brains, most PAX6+ RGCs were localized at the DNe, while TBR2+ IPCs were more evenly distributed along the DNe-FDJ-DG migratory/differentiating path (Fig. 4a–c). Although total numbers of PAX6+ RGCs, TBR2+ IPCs, and the ratio of dividing (PAX6+EdU+/PAX6+) and IPC producing RGCs (PAX6+TBR2+/PAX6+) were not significantly altered in cKO hippocampi (Fig. 4d–e, f, h), the ratio of dividing IPCs was decreased by 36.6% (Fig. 4g). The distribution of progenitors along the DNe-FDJ-DG path was dramatically shifted toward the DNe: the cKO DNes were significantly enriched with more PAX6+, PAX6+EdU+ and TBR2+ cells; whereas their distribution in cKO DGs was drastically reduced (Fig. 4i–p). Although the differentiating rate from RGCs to IPCs (PAX6+TBR2+/PAX6+) at FDJ is 60% higher in cKOs (Fig. 4p), numbers and the ratio of dividing TBR2+ cells at FDJ decreased by 56.3% and 57.0% respectively (Fig. 4l, n). By P7, most TBR2+ IPCs located at the DG and actively divide in control brains. However, numbers of total and dividing TBR2+ IPCs were significantly decreased at cKO DGs, whereas those at hippocampal SVZ and FDJ were greatly increased (Fig. 4q–u; Supplementary Fig. 7a). In summary, the migratory and differentiating trajectory of hippocampal progenitors, particularly IPCs, were greatly delayed, both spatially and temporally, upon loss of KDM2B-CxxC.

We further analyzed phenotypes in $Kdm2b^{Emx1-\Delta CxxC}$ cKO CAs. First, EdU was administered at E14.5 followed by phenotypic analyses at E18.5 (Supplementary Fig. 7b). In cKO CA1 region, significantly more EdU+ cells resided at VZ and fewer EdU+ cells located at pyramidal cell layer (Supplementary Fig. 7c). Moreover, 87.3% more EdU-labeled cells expressed TBR2 (Supplementary Fig. 7d). Second, in E16.5 cKOs, 41.7% and 18.4% more PAX6+ and TBR2+ cells were detected at the hippocampal neuroepithelia (HNE), where CA neurogenesis originates. While 35.7% more PAX6+ cells were proliferative (PAX6+EdU+), a smaller portion of TBR2+ cells were dividing (statistically not significant, Supplementary Fig. 7e–n). Thus, loss of KDM2B-CxxC also leads to hampered neurogenesis and neuronal migration at the CA region.

We next asked whether neuronal maturation was impeded on loss of KDM2B-CxxC. To this end, we crossed the $Kdm2b^{flox(CxxC)}$ mice with $Nex$-$Cre$ mice to obtain $Kdm2b^{Nex-\Delta CxxC}$ cKO mice, where KDM2B-CxxC was specifically ablated in all postmitotic neurons but not progenitors

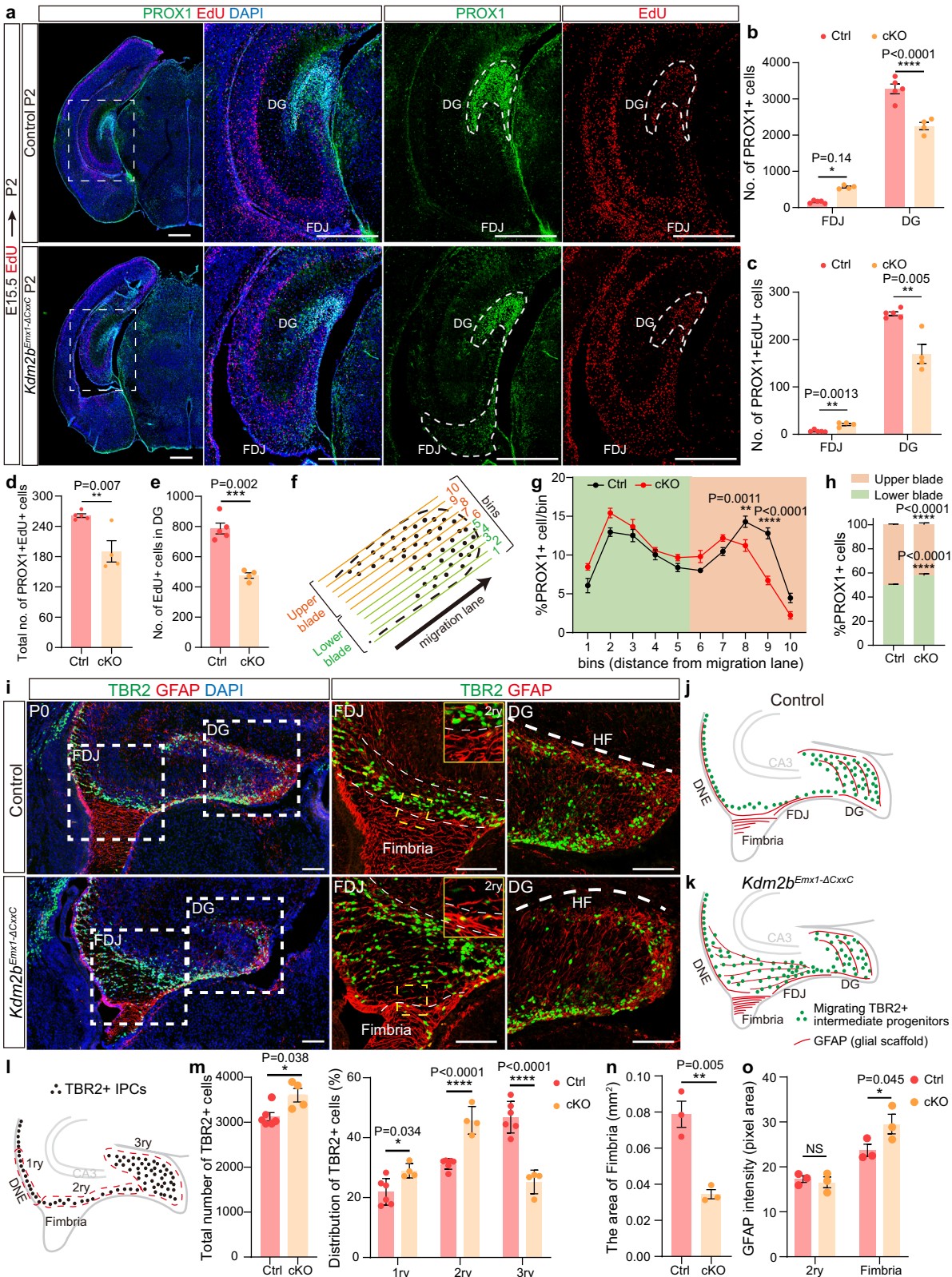

of neocortices and hippocampi. Phenotypic analyses revealed that hippocampal morphology and cellular components, including PROX1+ DG granule cells, of P7 *Kdm2b^Nex-ΔCxxC* cKOs did not show any abnormality, suggesting that hippocampal hypoplasia was not due to defects on postmitotic neuron differentiation (Supplementary Fig. 8a). Together, KDM2B regulates hippocampal morphogenesis by controlling multiple behaviors, including coordinated RGC to IPC

differentiation, migration, and divisions of neural progenitors (Supplementary Fig. 8b, c).

## Kdm2b-ΔCxxC aberrantly activates Wnt signaling in the HP

We next sought to unveil molecular events and mechanisms underlying hippocampal hypoplasia caused by KDM2B mutation. First, hippocampal tissue from P0 control and *Kdm2b^Emx1-ΔCxxC* cKO brains were

**Fig. 3 | Ablation of the KDM2B-CxxC impedes the migration of intermediate progenitors and production of granule cells. a** EdU was administered at E15.5 and double labeling of PROX1 and EdU was performed on P2 coronal sections. Dashed lines indicate DG and FDJ. **b, c** Quantification of PROX1+ (**b**) and PROX1+-EdU+ (**c**) cells in the FDJ and DG. **d** Quantification of total PROX1+EdU+ cells in hippocampi. **e** Quantification of EdU+ cells in the DG. **f** The 3ry matrix was divided into 10 ventral- -to-dorsal bins spanning the lower to upper blade domain. **g** The percentage of PROX1+ cells in each bin is represented. **h** Quantification of the percentage of PROX1+ granule neurons positioned in the lower blade (bins 1–5), versus the upper blade (bins 6–10) in P2 control and *Kdm2b*^Emx1-ΔCxxC DGs. **i** Double immunofluorescence of TBR2 (green) and GFAP (red) on P0 wild-type and *Kdm2b*^Emx1-ΔCxxC hippocampi. Boxed regions of FDJ and DG were enlarged on the right. **j, k** The schematic of P0 wild-type and *Kdm2b*^Emx1-ΔCxxC hippocampi.

**l, m** Distribution of TBR2+ cells along the three matrices, where dashed lines demarcate regions designated as the 1ry, 2ry, and 3ry matrix (**l**). *n* = 6 for control brains and n = 4 for *Kdm2b*^Emx1-ΔCxxC brains. **n** Quantification of the area of fimbria. **o** Quantification of GFAP immunofluorescence (pixel area) in the 2ry and fimbria. Cartoons in (**f**), (**j**) and (**l**) are adapted from Caramello et al.[36]. *n* = 5 for control brains and *n* = 4 for *Kdm2b*^Emx1-ΔCxxC brains (**a–h**). *n* = 3 for control brains and *n* = 3 for *Kdm2b*^Emx1-ΔCxxC brains (**n–o**). Data are represented as means ± SEM. Statistical significance was determined using an unpaired two-tailed Student's *t*-test (**d, e, m** left, **n**), or using two-way ANOVA followed by Sidak's multiple comparisons test (**b, c, g, h, m** right, **o**). *$P < 0.05$, **$P < 0.01$, ***$P < 0.001$, and ****$P < 0.0001$. Scale bars, 500 μm (**a**), 100 μm (**i**). DG dentate gyrus, DMS dentate migratory stream, FDJ fimbriodentate junction, HF hippocampal fissure, 1ry primary matrix, 2ry secondary matrix, 3ry tertiary matrix.

---

harvested and subjected to RNA-seq transcriptome analyses. We chose P0 because the third germinal primordium of the dentate gyrus just appears, and the hippocampal neurogenesis is at peak. Importantly, the blockade of IPC migration is prominent in P0 cKO hippocampi. Data showed 886 genes and 575 genes were activated and repressed respectively in *Kdm2b*^Emx1-ΔCxxC cKO hippocampi. In line with aforementioned phenotypic analyses, expression levels of markers for progenitor cells including *Pax6*, *Neurog2*, and *TBR2/Eomes* were increased in cKO hippocampi (Fig. 5a). Gene ontology (GO) analyses revealed that upregulated genes were involved in pattern formation, morphogenesis, cell proliferation and canonical Wnt signal pathways (Fig. 5b, Supplementary Fig. 9a), whereas down-regulated genes encompassing those associated with neuronal structures and functions (Fig. 5c). Notably, a series of canonical Wnt pathway components, including ligands, receptors, and signal transducers, were significantly activated upon loss of KDM2B-CxxC (Fig. 5d–f). To validate whether the canonical Wnt signaling was enhanced in cKO hippocampi, *Kdm2b*^Emx1-ΔCxxC cKO mice were crossed with the BAT-GAL [B6.Cg-Tg(BAT-lacZ)3Picc/J] Wnt-reporter mice. Beta-galactosidase staining showed that P0 cKO hippocampi had stronger canonical Wnt activity, including the CA region and FDJ. The ventricular surface of hippocampi and neocortices of cKO brains also displayed elevated Wnt signaling (Fig. 5g–i). In particular, components of canonical Wnt signaling genes such as *Lef1* and *Sfrp2* were elevated. In situ hybridization verified that the expression of *Lef1*, the gene encoding the transcriptional co-factor of β-Catenin to activate Wnt signaling, is significantly upregulated in E16.5 cKO hippocampi (Fig. 5j–l). LEF1 not only has an early role in specifying the hippocampus, but also controls the generation of dentate gyrus granule cells[37]. Similarly, the expression of *Sfrp2*, another Wnt signaling pathway component was greatly enhanced in E16.5 cKO hippocampi and VZ/SVZ of neocortices (Fig. 5m–o). Although members of the SFRP family were first reported as Wnt inhibitors, *Sfrp2* regulates anteroposterior axis elongation, optic nerve development, and cardiovascular and metabolic processes by the promoting or inhibiting Wnt signaling pathway[38–41].

Since P0 hippocampi contained multiple cell types ranging from RGCs to IPCs to neurons, we propagated RGCs in vitro under the serum-free neurosphere condition (Supplementary Fig. 9b). RNA-seq transcriptome studies revealed that a significant portion of activated and repressed genes in cKO neurospheres overlapped with those in cKO hippocampal tissue (Supplementary Fig. 9c–e). GO analyses indicated genes involved in cell division and canonical Wnt signaling were activated in cKO hippocampal neurospheres (Supplementary Fig. 9f). The canonical Wnt pathway genes *Lef1* and *Sfrp2* were also significantly upregulated in neurospheres derived from cKO hippocampi (Supplementary Fig. 9g).

KDM2B is a key component of variant PRC1 to mediate repressive histone modification H2AK119ub and subsequent H3K27me3 (Fig. 6a) and its long isoform KDM2BLF also bears demethylase activity for H3K36me2. We then examined how loss of KDM2B-CxxC affects these modifications in hippocampal tissue and neurospheres, and whether

these effects were associated with enhanced Wnt signaling. To this end, chromatin immunoprecipitation sequencing (ChIP-seq) was performed using P0 hippocampal tissue and neurospheres. As expected, the overall levels of H2AK119ub and H3K27me3 were decreased in cKO hippocampal tissues and neurospheres (Fig. 6b, c). However, the level of H3K36me2 was not altered in cKO tissue but slightly increased in neurospheres (Fig. 6b, c; Supplementary Fig. 10g–h). Consistently, assay for transposase-accessible chromatin using sequencing (ATAC-seq) showed that chromatin became more accessible in cKO hippocampi (Fig. 6c).

We next verified changes of Wnt signaling pathway and investigated how histone modifications and chromatin accessibilities correlate with changes of gene expression. Quantitative reverse transcription PCR (RT-qPCR) revealed that the transcripts of multiple Wnt ligands and *Sfrp2* in cKO hippocampi were significantly elevated throughout developmental time points (Fig. 6d–g; Supplementary Fig. 10a–f). Since PRC1.1 could be recruited to CGIs *via* KDM2B's CxxC zinc finger to catalyze H2AK119ub, we paid special attention to chromatin status of CGIs. Peaks for H2AK119ub and H3K27me3 were indeed decreased on CGI-enriched promoters of many activated genes, including *Wnt7b*, *Wnt3*, *Wnt6*, *Wnt10a* and *Sfrp2*, as well as *Pax6*, *Eomes* and *Neurod1*; but peaks for H3K36me2 on these sites were not significantly altered in P0 cKO hippocampi (Fig. 6h; Supplementary Fig. 10i). Consistently, ATAC-seq showed CGI promoters of many de-repressed genes were more accessible in cKO hippocampi, such as *Wnt7b Wnt3*, *Sfrp2*, *Eomes* and *Neurod1* (Fig. 6h; Supplementary Fig. 10i). For *Lef1*, the enrichment of H2AK119ub and H3K27me3 was diminished around its CGI promoter in cKO neurospheres (Supplementary Fig. 10j).

To ask whether activating the Wnt pathway could hamper hippocampal neurogenesis, we electroporated a mix of plasmids expressing Wnt ligands (Wnt3a, Wnt5a, Wnt5b, Wnt7b, and Wnt8b) into E14.5 hippocampal primordia. Since *Sfrp2* is one of the most enhanced Wnt signaling components upon loss of KDM2B-CxxC, constructs overexpressing *Sfrp2* were separated transduced into E14.5 hippocampal primordia (Fig. 7a–d). Data showed that overexpressing Wnt ligands or *Sfrp2* could significantly block hippocampal neurogenesis, as more transduced cells resided in VZ and IZ but fewer in the pyramidal cell layer (Py) compared to controls (Fig. 7e, g), with significantly more transduced cell co-expressing TBR2 (Fig. 7i). Consistently, significantly fewer electroporated GFP+ cells reached the distal (3ry) region of DG (Fig. 7f, h), while more electroporated GFP+ cells expressed TBR2 at DGs (Fig. 7j).

## KDM2B selectively suppresses the Wnt signaling in the HP

During early brain development, patterning of hippocampus is majorly controlled by Wnt and BMP signaling from hem while patterning of neocortex is regulated by signals from anti-hem, with the pallial-subpallial boundary (PSPB) being the signal center[42, 43]. We collected hippocampal and neocortical tissues of P0 control and *Kdm2b*^Emx1-ΔCxxC brains for RNA-seq analysis (Supplementary Fig. 11a–g). First, Wnt

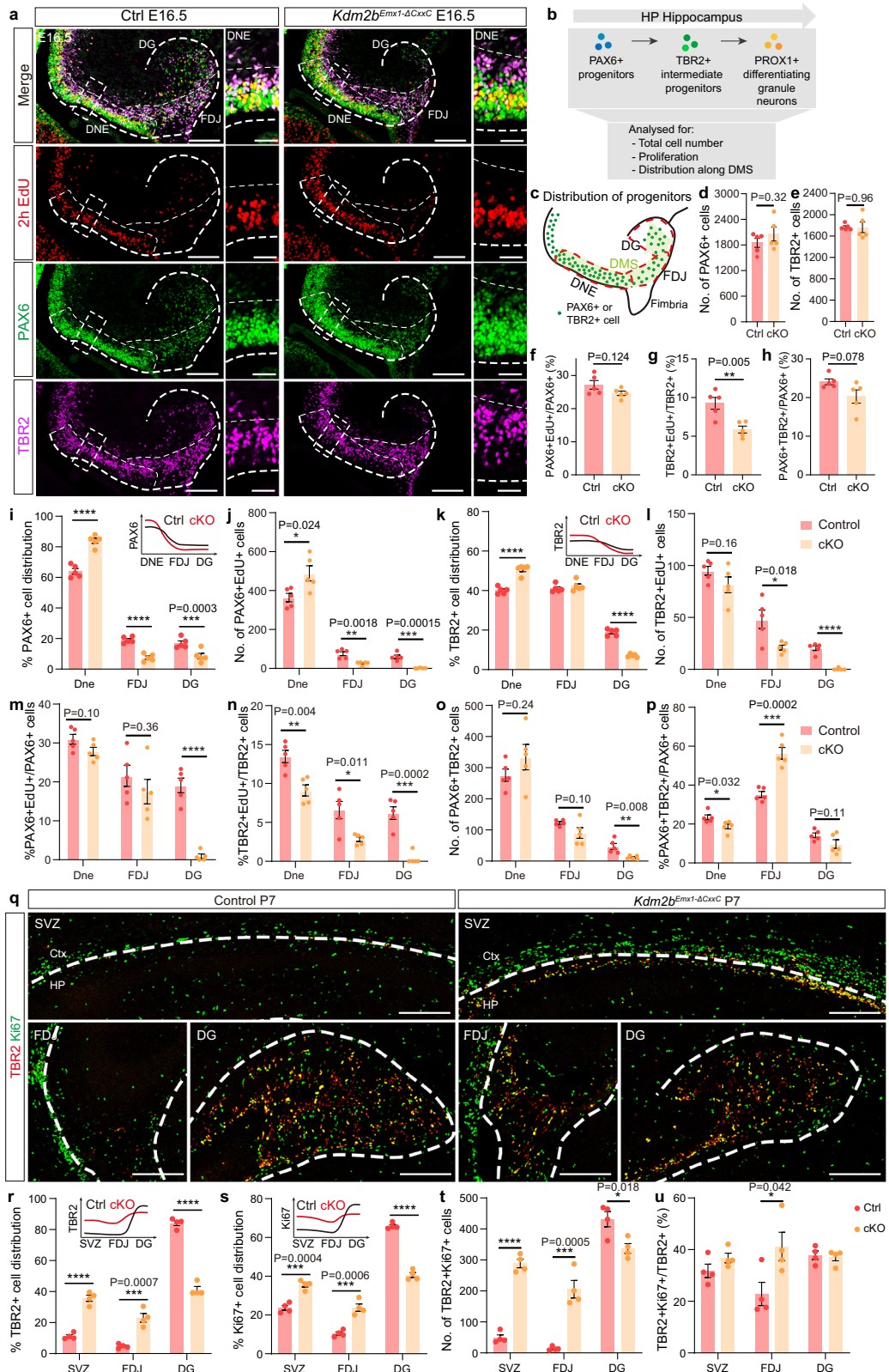

signal genes are more abundant in the hippocampus, while PSPB genes, including *Pax6* and *Sfrp2*, are more abundant in the neocortex (Supplementary Fig. 11a, f), suggesting that the two signal gradients persist through perinatal stage and the hem signal continues to affect hippocampal neurogenesis but exert little effect on neocortex. Secondly, both hem genes and PSPB genes were significantly upregulated in *Kdm2b^{Emx1-ΔCxxC}* hippocampi. In contrast, the change of Wnt signal genes in *Kdm2b^{Emx1-ΔCxxC}* neocortex was not obvious, while PSPB signal genes in neocortex were mildly upregulated (Supplementary Fig. 11a–e). Hence, the silence of Wnt signaling by KDM2B is more dominant in hippocampus than in neocortex, which could explain why neocortical development is not disturbed by the loss of KDM2B-CxxC.

**Fig. 4 | Blocked differentiation of neural progenitors on loss of KDM2B-CxxC.** **a** Triple-labeling of PAX6 (green), TBR2 (violet) and EdU (red, labeled 2 h) on E16.5 control and *Kdm2b*[Emx1-ΔCxxC] brain sections. Boxed regions were enlarged on the right, and single channel fluorescence images of PAX6, TBR2 and EdU were shown respectively. Dashed lines outline the hippocampi and distinguish DNE, FDJ and DG. **b** Experimental analysis scheme. The cartoon in Fig. 4b is adapted from Caramello et al.[36]. **c** The schematic of E16.5 wild-type hippocampi. Green dots represent PAX6+ progenitors or migrating TBR2+ intermediate progenitors. **d, e** Quantification of total PAX6+ and TBR2+ cells. **f–h** Quantification of the proportion of PAX6+EdU+/PAX6+ (**f**), TBR2+EdU+/TBR2+ (**g**) and PAX6 + TBR2+/PAX6+ (**h**). **i–p** Quantification of the distribution of PAX6+ (**i**), PAX6+EdU+ (**j**), TBR2+ (**k**), TBR2+EdU+ (**l**) and PAX6 + TBR2+ cells (**o**), and quantification of the proportion of PAX6+EdU+/PAX6+ (**m**), TBR2+EdU+/TBR2+ (**n**) and PAX6 + TBR2+/PAX6+ (**p**) along the DMS. The distribution patterns of PAX6+ (**i**) and TBR2+ (**k**) cells in control

and cKO hippocampi were shown as line graphs on the upper right corners. **q** Double-labeling of TBR2 (red) and Ki67 (green) on P7 coronal section of control and *Kdm2b*[Emx1-ΔCxxC] SVZ, FDJ and DG. Dashed lines outline the hippocampi. Immunofluorescence staining of whole brain sections were shown in Supplementary Fig. 7a. **r–u** Quantification of the distribution of TBR2+ (**r**), Ki67+ (**s**) and TBR2 + Ki67+ cells (**t**), and the proportion of TBR2 + Ki67+/TBR2+ (**u**) at SVZ, FDJ and DG. *n* = 5 for control brains and *n* = 5 for *Kdm2b*[Emx1-ΔCxxC] brains (**d–p**). *n* = 4 for control brains and *n* = 4 for *Kdm2b*[Emx1-ΔCxxC] brains (**r–u**). Data are represented as means ± SEM. Statistical significance was determined using an unpaired two-tailed Student's *t*-test (**d–h**), or using two-way ANOVA followed by Sidak's multiple comparisons test (**i–p, r–u**). *$P < 0.05$, **$P < 0.01$, ***$P < 0.001$, and ****$P < 0.0001$. Scale bars, 200 µm (**a** left, **q**), 50 µm (**a** right). DNE dentate neuroepithelium, FDJ fimbriodentate junction, DG Dentate Gyrus, SVZ subependymal ventricular zone, Ctx Cortex, HP Hippocampus.

## Loss of Ring1B did not cause accumulation of progenitors

KDM2B recruits other component of PRC1.1, including the ubiquitin protein ligase Ring1B, to CGIs to initiate and stabilize gene silencing. We then asked whether the impeded migration and differentiation of neural progenitors in *Kdm2b* cKO hippocampi is caused by PRC1's loss-of-function. We obtained *Rnf2* (the gene encoding Ring1B) cKO mice—*Rnf2*[Emx1-cKO]—by crossing floxed *Rnf2* mice with *Emx1*-Cre mice (Supplementary Fig. 11h). As expected, ablation of *Rnf2* greatly decreased the level of H2AK119ub in P0 neocortical tissues, with levels of H3K27me3 also slightly decreased (Supplementary Fig. 11i). We then stained P0 brains with TBR2 (Supplementary Fig. 11j, k) to find *Rnf2*[Emx1-cKO] hippocampi were smaller than controls and the number of TBR2+ progenitors in *Rnf2*[Emx1-cKO] hippocampi was decreased by 9.2%. Moreover, the distribution of TBR2+ progenitors in DGs, but not DNE and FDJ, was significantly decreased in the *Rnf2*[Emx1-cKO] hippocampi. However, to our surprise, there was no accumulation and dispersion of TBR2+ IPCs in the FDJ region (2ry) of the *Rnf2*[Emx1-cKO] hippocampi as found in *Kdm2b*[Emx1-ΔCxxC] cKO brains (Supplementary Fig. 11l). Therefore, although PRC1's loss-of-function also causes hippocampal hypoplasia, it did not lead to buildup of neural progenitors in the migrating path of developing hippocampi. Transcriptome analyses showed that although Wnt pathway and neurogenesis genes were upregulated in *Kdm2b*[Emx1-ΔCxxC] hippocampi, they were not significantly altered in *Rnf2*[Emx1-cKO] hippocampal tissue (Supplementary Fig. 11m). Thus, insufficient PRC1 activity alone does not account for de-repressed Wnt signaling and subsequent defects of IPC fates. KDM2B could selectively mediate deposition of repressive histone marks on many Wnt signal genes in the developing hippocampi. Furthermore, expression alterations of other targets of Ring1B and PRC1 might counter some effects caused by loss of KDM2B-CxxC.

Together, loss of KDM2B-CxxC reduces repressive histone modifications on key Wnt signal genes, hence leading to prolonged Wnt activation, which causes hampered differentiation and migration of hippocampal progenitors (Fig. 7k).

## Discussion

The hippocampus is evolutionarily more ancient than the neocortex[44] and the production and localization of CA pyramidal neurons also follows the birthdate-dependent inside-out pattern[45]. In neocortical development, Pax6+ RGCs and TBR2+ IPCs largely reside in the VZ and SVZ respectively, with their nuclei undergoing local oscillation. Although many cellular and epigenetic programs were found to control numbers and differentiation of neocortical RGCs and IPCs, it remains unclear how these mechanisms were applied in hippocampal development, which involves migration and dispersion of neural progenitors. Here we revealed that the chromatin association of KDM2B, an essential component of variant PRC1.1, is required for hippocampal formation. KDM2B mediates silencing of Wnt signaling genes to facilitate proper migration and differentiation of hippocampal progenitors.

Our knowledge on the mammalian Polycomb repressive system has mostly come from studies in pluripotent stem cells and in embryos at early developmental stages[6,15,46–48], when the establishment of repressive domains is initiated[49]. Nonetheless, how Polycomb controls sequential fate determination in specific tissues and at later developmental stages largely remains elusive. Both PRC1 and PRC2 are essential players in neocortical development[50–54]. The deletion of Ring1B, the core enzymatic component of PRC1, prolonged neocortical neurogenesis at the expense of gliogenesis. PRC1 regulates the chromatin status of neurogenic genes of neural progenitors, hence altering their responsiveness to neurogenic Wnt signals over developmental time[55]. Ring1B was also found to regulate dorsoventral patterning of the forebrain[51] and sequential production of deep and upper-layer neocortical PNs[52]. However, we surprisingly revealed that ablation of Ring1B did not significantly hamper migration and distribution of TBR2+ neural progenitors of developing hippocampi, whereas these defects are prominent in the *Kdm2b*[Emx1-ΔCxxC] hippocampi. In addition, removal of KDM2B from the chromatin only have mild effects on neocortical development and overall H2AK119ub1 and H3K27me3 levels. Therefore, KDM2B likely controls fate determination of hippocampal progenitors by selectively repressing a series of progenitor genes including those in the Wnt pathway *via* PRC1.1. How KDM2B selectively targets these genes during hippocampal morphogenesis remains to be investigated.

Knocking out EED, one of the core components of PRC2, results in hampered neurogenesis of postnatal DG. However, unlike *Kdm2b* cKO brains, the EED knockouts did not display any hippocampal malformation at P0[20]. Moreover, deletion of *Ezh2* in adult NSCs leads to disturbed neurogenesis of DG[21], which was unseen in *Kdm2b* cKO DGs. These discrepancies echoes either distinct roles or spatiotemporal activities of PRC1 and PRC2. It would be worthy of dissecting functions of distinct PRC1/2 variants or their components in neural development and homeostasis[56]. A number of studies indicated that PRC1 and PRC2 can directly or indirectly regulate Wnt signaling in developmental, physiological, and disease conditions[57–59]. Wnt signaling governs multiple aspects of neural development including neurulation, pattern formation, and fate choices of neural progenitors[60–62]. Moreover, the strength and gradient of the canonical Wnt signaling in the RGC-IPC-neuron path and through developmental time ensures proper cell fate establishment and transition[63–65], including those in hippocampal morphogenesis[4,37]. Deletion of KDM2B-CxxC greatly elevated Wnt signaling in hippocampi of multiple developing stages and in hippocampal progenitors, which could lead to impeded migration and differentiation of IPCs. Of note, however, the ablation of KDM2B-CxxC has minimal effects on neocortical development, reflecting regional and timing difference between neocortical and hippocampal progenitors. For instance, at P0 stage, when hippocampal neurogenesis is at the peak while neocortical neurogenesis is turned off, Wnt signaling in hippocampus is stronger than that in neocortex (Fig. 5g, h, Supplementary Fig. 11a). Sustained Wnt activation on loss of KDM2B hence

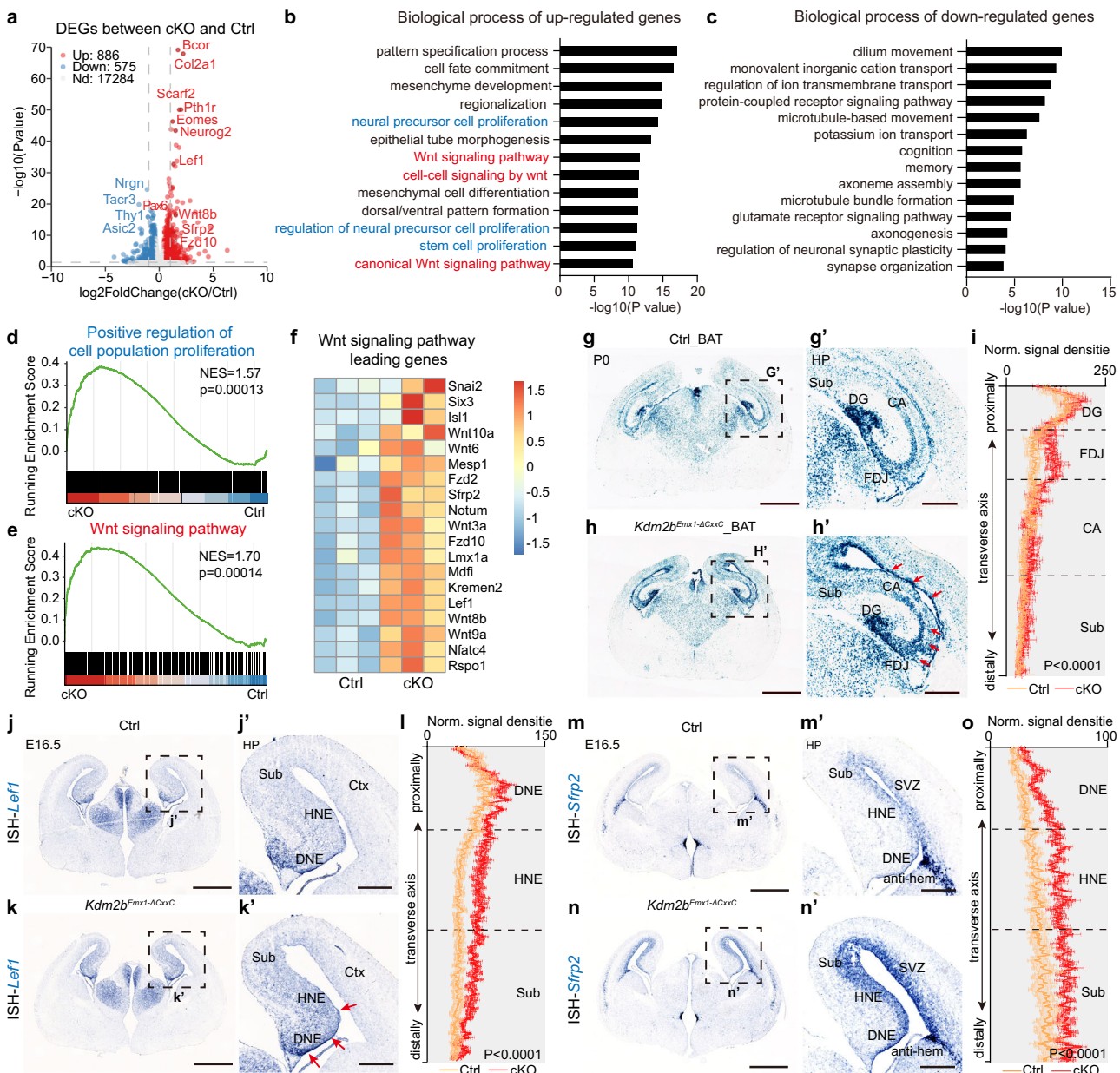

**Fig. 5 | Loss of KDM2B-CxxC results in activation of the Wnt signaling pathway in the hippocampus. a** The volcano plot of genes upregulated (red) and down-regulated (blue) in P0 *Kdm2b^{Emx1-ΔCxxC}* hippocampi compared to controls, analyzed using DESeq2 (version 1.38.1). **b, c** GO analysis of the biological process of upregulated (**b**) and down-regulated (**c**) genes in P0 *Kdm2b^{Emx1-ΔCxxC}* hippocampi, revealed terms related to cell proliferation (blue) and Wnt signaling pathways (red). **d, e** GSEA analysis of positive regulation of cell population proliferation (**d**) and Wnt signaling pathway (**e**). **f** The heat map of leading genes in (**e**). **g, h** X-gal staining on P0 coronal sections of Control_BAT (**g**) and *Kdm2b^{Emx1-ΔCxxC}*_BAT (**h**) brains. Boxed regions were enlarged on the right (**g'** and **h'**). Red arrows indicate areas where X-gal signals are significantly enhanced. **i** Quantification of normalized X-gal signal density on the DG-CA-FDJ-Sub path. *n* = 4 for Control_BAT brains and *n* = 3 for *Kdm2b^{Emx1-ΔCxxC}*_BAT brains. **j, k** In situ hybridization (ISH) of *Lef1* on E16.5 Control (**j**) and *Kdm2b^{Emx1-ΔCxxC}* (**k**) coronal brain sections, with boxed regions magnified on the

right (**j'** and **k'**). Red arrows indicate areas where the *Lef1* expression is significantly elevated. **l** Quantification of normalized ISH signal density of *Lef1* on the DNE-HNE-Sub path. *n* = 4 for control brains and *n* = 4 for *Kdm2b^{Emx1-ΔCxxC}* brains. **m, n** ISH of *Sfrp2* on E16.5 control (**m**) and *Kdm2b^{Emx1-ΔCxxC}* (**n**) coronal brain sections, with boxed regions magnified on the right (**m'** and **n'**). **o** Quantification of normalized ISH signal density of *Sfrp2* on the DNE-HNE-Sub path. *n* = 4 for control brains and *n* = 4 for *Kdm2b^{Emx1-ΔCxxC}* brains. Gene Ontology (GO) analysis and Gene Set Enrichment Analysis (GSEA) in (**b**–**e**) were performed using clusterProfiler (version 4.2.2). Data are represented as means ± SEM. Statistical significance was determined using two-way ANOVA multiple comparisons test (**i**, **l**, **o**). Scale bars, 1 mm (**g**, **h**, **j**, **k**, **m** and **n**), 300 μm (**g'**, **h'**, **j'**, **k'**, **m'** and **n'**). HP Hippocampus, DG dentate gyrus, FDJ fimbriodentate junction, CA Cornu Ammonis, Sub Subiculum, DNE dentate neuroepithelium, HNE hippocampal neuroepithelium, Ctx cortex, SVZ subependymal ventricular zone.

could have greater impact on hippocampal neurogenesis compared to neocortical development. Interestingly, deletion of KDM2B-CxxC in adult neural stem cells exerts no effect on DG neurogenesis, probably reflecting transient expression of KDM2B in embryonic and postnatal hippocampi.

The deletion of KDM2B-CxxC would totally abolish KDM2B's association with CGIs, thus disabling KDM2B's two major functions on chromatin - mediating H2AK119Ub *via* PRC1.1 and demethylating H3K36me2, the latter of which is solely executed by the JmjC-containing KDM2BLF. Previous studies indicated that the

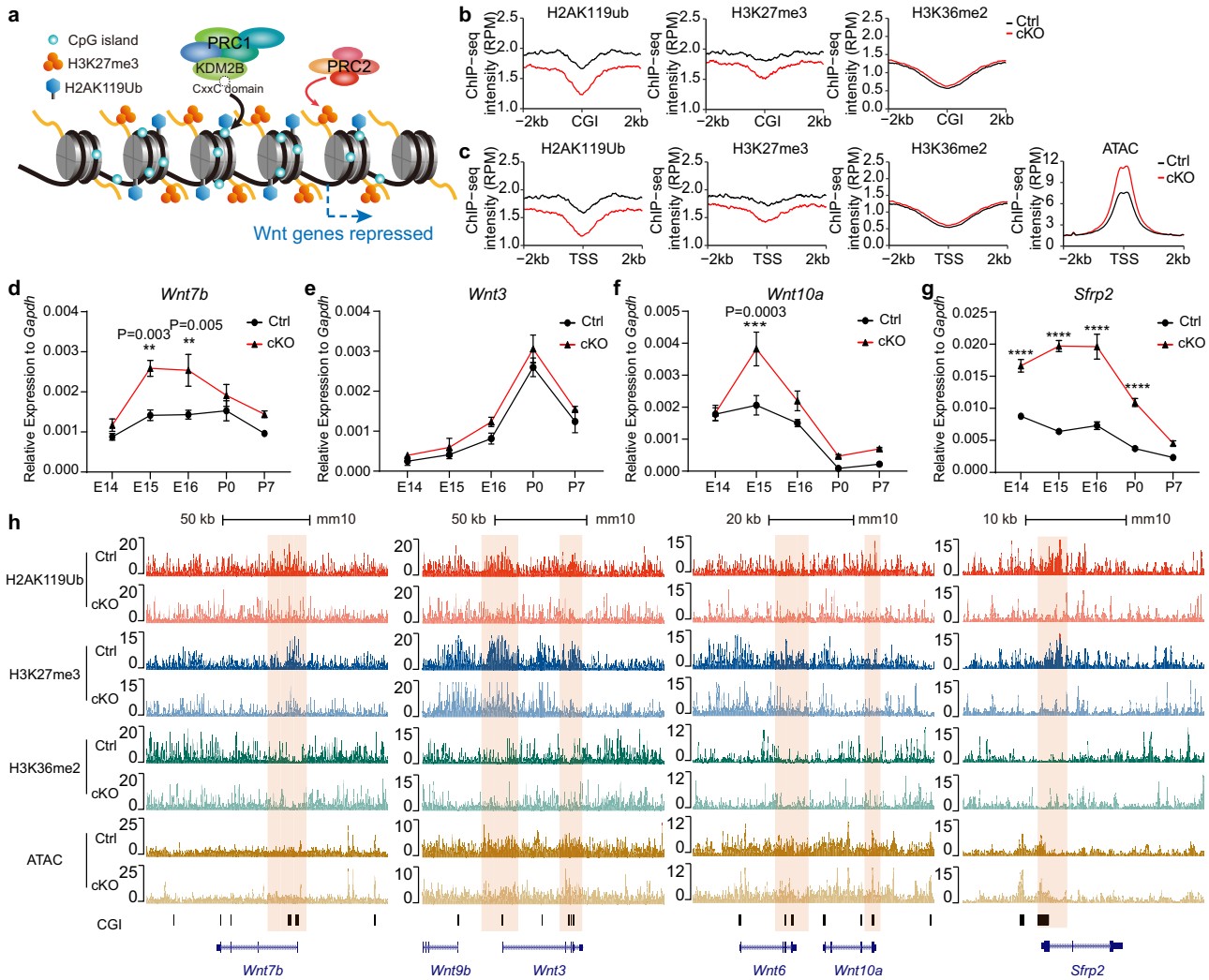

**Fig. 6 | KDM2B epigenetically silences components of Wnt signaling genes in developing hippocampi. a** The working diagram of KDM2B: KDM2B-CxxC recognizes and binds to CpG islands (CGI) of DNA, therefore recruiting the PRC1 to CpG islands (CGIs). Reciprocal recognition of modifications by PRC1 and PRC2 leads to enrichment of H2AK119ub and H3K27me3, hence stabilizing gene repression. **b** Line charts showing average H2AK119ub, H3K27me3 and H3K36me2 signals at CGIs (±2 kb flanking regions) in P0 control (black lines) and *Kdm2b^{Emx1-ΔCxxC}* (cKO) (red lines) hippocampi. **c** Line charts showing average H2AK119ub, H3K27me3, H3K36me2 and ATAC-seq signals at CGI + TSS (±2 kb flanking regions) in P0 control (black lines) and *Kdm2b^{Emx1-ΔCxxC}* (cKO) (red lines) hippocampi. TSS transcription starting sites. **d–g** RT-qPCR showing relative expressions of *Wnt7b*, *Wnt3*, *Wnt10a* and *Sfrp2* in control (black lines) and *Kdm2b^{Emx1-ΔCxxC}* (red lines) hippocampi of indicated developmental stages (E14.5, E15.5, E16.5, P0 and P7). **h** The UCSC genome browser view of HA2K119ub, H3K27me3, and H3K36me2 enrichment and ATAC-seq signal in P0 control and *Kdm2b^{Emx1-ΔCxxC}* (cKO) hippocampi at Wnt gene loci [corresponding to (**d, f, g**), *Wnt7b*, *Wnt3*, *Wnt10a* and *Sfrp2*]. CGIs were shown as black columns at the bottom, and signals represent ChIP-seq RPM (reads per million). Colored regions marked enrichment differences between control and cKO. For E14.5, *n* = 4 for control brains and *n* = 4 for *Kdm2b^{Emx1-ΔCxxC}* brains; for E15.5, *n* = 3 for control brains and *n* = 3 for *Kdm2b^{Emx1-ΔCxxC}* brains; for E16.5, *n* = 3 for control brains and *n* = 3 for *Kdm2b^{Emx1-ΔCxxC}* brains; for P0, *n* = 2 for control brains and *n* = 4 for *Kdm2b^{Emx1-ΔCxxC}* brains; for P7, *n* = 3 for control brains and *n* = 3 for *Kdm2b^{Emx1-ΔCxxC}* brains (**d–g**). Data are represented as means ± SEM. Statistical significance was determined using two-way ANOVA followed by Sidak's multiple comparisons test (**d–g**). *$P < 0.05$, **$P < 0.01$, ***$P < 0.001$, and ****$P < 0.0001$.

demethylase activity of KDM2A/B is required for PRC establishment at CGIs of peri-implantation embryos[24], but contributes moderately to the H3K36me2 state at CGI-associated promoters and is dispensable for normal gene expression in mouse embryonic stem cells[66]. Interestingly, levels of H3K36me2 were not significantly altered globally or locally in KDM2B-CxxC deleted hippocampi. Moreover, *Kdm2b^{fKO}* mice did not display hippocampal hypoplasia or malformation[29,67]. Thus, the H3K36me2 demethylase activity of KDM2B is likely dispensable for hippocampal development.

*KDM2B* is implicated in neurological disorders including ID and behavior abnormalities, and the region encoding the CxxC ZF is the mutational hotspot[27]. Consistently, *Kdm2b^{Emx1-ΔCxxC}* cKO mice displayed prominent defects in spatial and motor learning and memory, as well

as contextual fear conditioning. It would be essential to explore whether patients with *KDM2B* mutations have defects of hippocampal morphogenesis and function, and how KDM2B-mediated gene repression is implicated in human brain development. More specific behavioral tests, such as novel object recognition and two-choice spatial discrimination test, could better describe how hippocampal functions were affected by KDM2B mutations.

## Methods

### Ethics

All experimental procedures were approved by the Animal Care and Ethical Committee of Medical Research Institute, Wuhan University. Informed consent was obtained from all participants of the study.

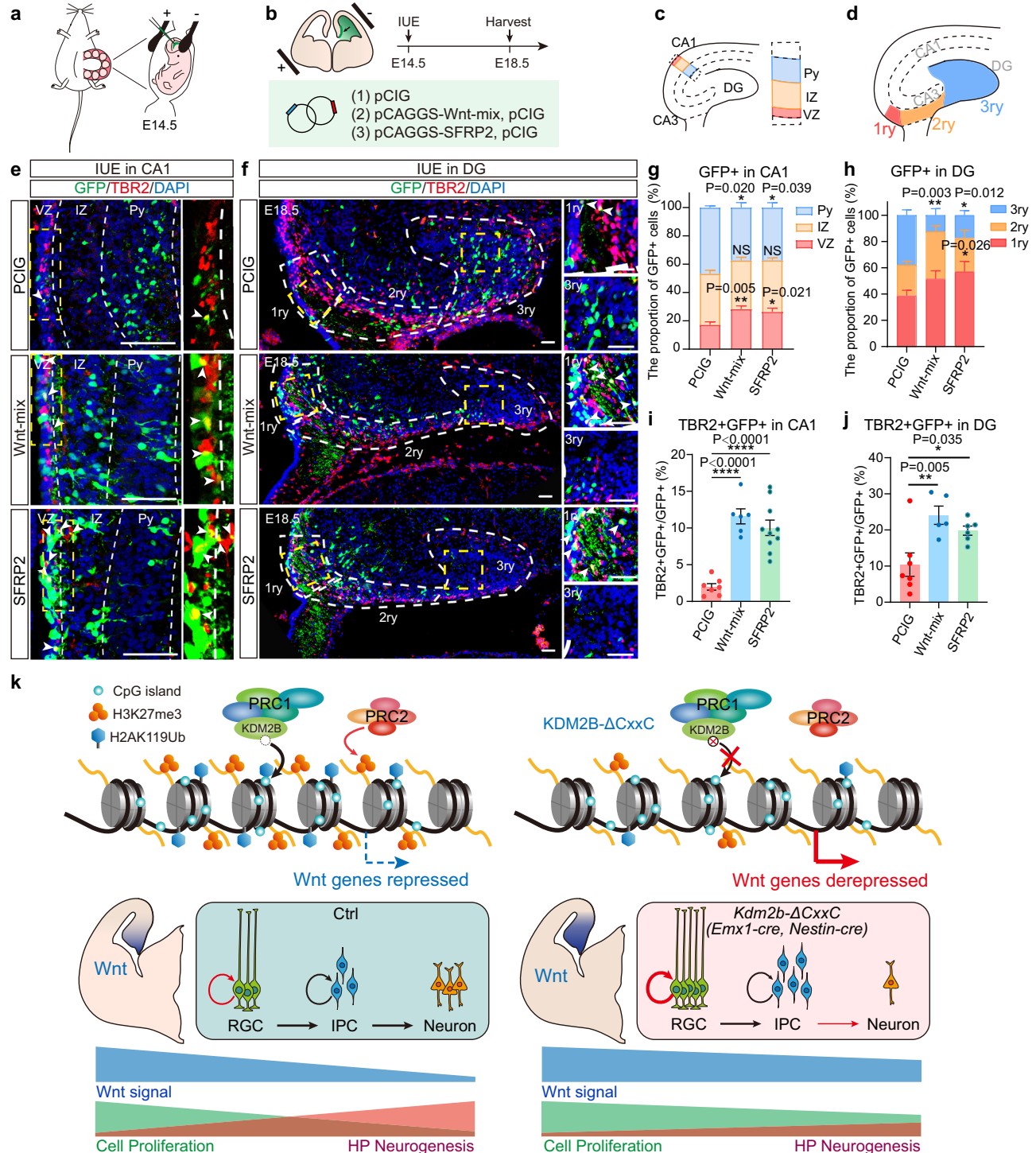

**Fig. 7 | Aberrant activation of the Wnt signaling in blocks hippocampal neurogenesis. a, b** The schematic diagram of *in utero* electroporation (IUE) to target the developing hippocampi. E14.5 mouse hippocampi were electroporated with empty or Wnt-mix-expressing vector (Wnt3a, Wnt5a, Wnt5b, Wnt7b, and Wnt8b) or SFRP2-expressing vector, along with the GFP-expressing vector (PCIG) to label transduced cells. Embryos were sacrificed at E18.5 for immunofluorescent analysis. **c, d** The schematic diagram of hippocampal structure, and the hierarchical partition of CA1 (VZ, IZ, Py) (**c**) and DG (1ry, 2ry, 3ry) (**d**). **e, f** Representative immunofluorescent images showing expression of TBR2+ (red) in GFP+ (green) transduced cells at E18.5 CA1 regions. Arrowheads denote double-labeled cells. White dashed lines distinguish three layers of CA1: VZ, IZ, Py in (**e**) and 1ry, 2ry, 3ry of DG (**f**). **g, h** The relative location of GFP+ cells in VZ, IZ, Py (**g**) and 1ry, 2ry, 3ry (**h**) were quantified. **i, j** Quantification of the proportion of TBR2 + GFP + /GFP+ in CA1(**i**) or in

DG (**j**) of PCIG, Wnt-mix and SFRP2. **k** Schematic diagram of KDM2B on hippocampal development: Loss of KDM2B-CxxC reduces repressive histone modifications−H2AK119ub and H3K27me3−on key Wnt signal genes, hence leading to prolonged Wnt activation over time. Hampered differentiation and migration of hippocampal progenitors leads to hippocampal hypoplasia. *n* = 7 for PCIG brains, *n* = 6 for Wnt-mix brains and *n* = 10 for SFRP2 brains in (**g**, **i**). *n* = 7 for PCIG brains, *n* = 5 for Wnt-mix brains and *n* = 6 for SFRP2 brains in (**h**, **j**). Data are represented as means ± SEM. Statistical significance was determined using two-way ANOVA followed by Tukey's multiple comparisons test (**g**, **h**), or using one-way ANOVA analysis (**i**, **j**). *$P < 0.05$, **$P < 0.01$, ***$P < 0.001$, and ****$P < 0.0001$. Scale bars, 50 μm (**e**, **f**). CA Cornu Ammonis, VZ ventricular zone, IZ intermediated zone, Py pyramidal cell layer of the hippocampus.

## Mice

Wild-type CD-1 (ICR) and C57BL/6 mice were obtained from the Hunan SJA Laboratory Animal Company (Changsha, China). Mice were housed in a certified specific-pathogen-free (SPF) facility. The noon of the day when the vaginal plug was found was counted as embryo (E) day 0.5.

Mouse lines used in this study included: $Kdm2b^{fl/fl}$ (generated by Applied Stem Cell), $Emx1$-$Cre$ (Jackson Laboratories, stock number 005628), $Nestin$-$Cre$ (Jackson Laboratories, stock number 003771), $Nex$-$Cre$ males [$Neurod6^{tm1(cre)Kan}$, MGI:2668659], $Nestin$-$CreERT2$ (Jackson Laboratories, stock number 016261), $Ai14$ (Rosa-CAG-LSL-tdTomato-WPRE), BAT-Gal mice (Jackson Lab, stock number 005317) and $Rnf2^{fl/fl}$ (purchased from GemPharmatech, Strain NO. T014803). Detailed mouse mating and genotyping methods are described in Supplementary Methods.

## Tamoxifen and BrdU administration

To activate Cre-mediated recombination, tamoxifen (TAM; Sigma-Aldrich) was used, which was made fresh daily and dissolved in sunflower oil solution (Sigma-Aldrich). 8-week-old $Kdm2b^{NestinCreERT2-\Delta CxxC}$ mice were daily administered with 30 mg/kg prewarmed TAM intraperitoneally for 6 consecutive days (d1-d6). From day 2 to day 7, mice were injected with 50 mg/kg BrdU (Sigma-Aldrich) intraperitoneally for 6 consecutive days and were sacrificed 1 day later (day 8, Short-Term) or 4 weeks later (day 35, Long-Term) to identify BrdU-positive adult-born cells.

## Tissue fixation and sectioning

The pregnant dam was anesthetized with 0.7% w/v pentobarbital sodium (105 mg/kg body weight) in 0.9% sodium chloride. Embryos were sequentially removed from the uterus. Brains of embryos were dissected out in cold PBS and immersed in 4% paraformaldehyde (PFA) overnight at 4 °C. For P0, P2, P7 and adult mice, animals were anesthetized with 0.7% w/v pentobarbital sodium solution followed by trans-cardiac perfusion with 4% PFA in PBS (P0, 5 ml; P7, 10 ml; adult, 30 ml). Brains were dissected and post-fixed in 4% PFA overnight at 4 °C. Next day, brains were dehydrated in 20% w/v sucrose overnight at 4 °C. For sectioning, brains were embedded in OCT (SAKURA) and cut at 20 μm for adult brains and 14 μm for other stages with a cryostat (Leica CM1950).

## Nissl staining

Adult brain sections were stained with 0.25% Cresyl Violet (Sigma-Aldrich) solution for 15 min at 65 °C. Sections were then decolorized in ethanol for 0.5–1 min, dehydrated in gradient ethanol solutions for 5 min each and cleared twice in xylene for 5 min. Sections were mounted in the neutral balsam.

## In situ hybridization (ISH)

Sections were dried in a hybridization oven at 50 °C for 15 min and fixed in 4% PFA for 20 min at room temperature, followed by permeabilization in 2 μg/ml proteinase K in PBS for 10 min at room temperature. Prior to hybridization, sections were acylated in 0.25% acetic anhydride for 10 min. Then, sections were incubated with a digoxigenin-labeled probe diluted (0.2 ng/μl) in hybridization buffer (50% deionized formamide, 5× SSC, 5× Denhart's, 250 μg/ml tRNA, and 500 μg/ml Herring sperm DNA) under coverslips in a hybridization oven overnight at 65 °C. The next day, sections were washed four times for 80 min in 0.1× SSC at 65 °C. Subsequently, they were treated with 20 μg/ml RNase A for 20 min at 37 °C, then blocked for 3.5 h at room temperature in 10% normal sheep serum. Slides were incubated with 1:5000 dilution of anti-digoxigenin-AP conjugated antibody (Roche) overnight at 4 °C. BCIP/NBT (Roche) was used as a color-developing agent. ISH primers used are listed in Supplementary Table 2.

## Immunofluorescence

Frozen brain sections were mounted onto Superfrost plus slides and then dried at room temperature. For heat-mediated antigen retrieval, slides were incubated for 15 min in 10 mM sodium citrate buffer (pH 6.0) at 95 °C. For BrdU staining, sections were treated with 20 μg/ml proteinase K (Sigma) (1:1000 in PBC) for 5 min and 2 N HCl for 30 min at room temperature. Sections were then immersed in blocking buffer (3% normal sheep serum and 0.1% Triton X-100 in PBS; or 5% BSA and 0.5% Triton X-100 in PBS) for 2 h at room temperature. Sections were then incubated in primary antibodies [mouse anti-Calbindin (1:1000; Sigma, C9848), rabbit anti-ZBTB20 (1:1000; Sigma, HPA016815), mouse anti-HopX (1:200; Santa Cruz, sc-398703), rabbit anti-Wfs1 (1:1000; Proteintech, 86995), mouse anti-PROX1 (1:200; Millipore, MAB5654), rabbit anti-GFAP (1:500; DAKO, Z0334), rat anti-CTIP2 (1:500; Abcam, ab18465), rabbit anti-SATB2 (1:500; Abcam, ab92446), rat anti-BrdU (1:500; Abcam, ab6326), mouse-anti-BrdU (1:500; Roche, 11170376001), rabbit-anti-DCX (1:500; Abcam, ab18723), rabbit anti-TBR2 (1:500; Abcam, ab23345), rat anti-TBR2 (1:500; Thermo Fisher, 14-4875-82), rabbit anti-PAX6 (1:500; Millipore, ab2237), and rabbit anti-Ki67 (1:500; Abcam, ab15580] in blocking buffer overnight at 4 °C. After three rinses in PBS, sections were incubated in secondary antibodies (Alexa Fluor 488-conjugated anti-mouse, A11029; Alexa Fluor 555-conjugated anti-mouse, A21422; Alexa Fluor 488-conjugated anti-rat, A11006; Alexa Fluor 555-conjugated anti-rat, A21434; Alexa Fluor 647-conjugated anti-rat, A21247; Alexa Fluor 488-conjugated anti-rabbit, A11034; Alexa Fluor 555-conjugated anti-rabbit, A21429; Alexa Fluor 647-conjugated anti-rabbit, A21245; Alexa Fluor 488-conjugated anti-chicken, A11039; Thermo Fisher Scientific; 1:1000) for 1 h at room temperature. Nuclei were labeled by incubation in PBS containing 4′,6-diamidino-2-phenylindole (DAPI) (0.1 μg/ml) (Sigma-Aldrich), and samples were mounted in ProLong Gold Antifade Mountant (Thermo Fisher Scientific).

## 5-Ethynyl-2′-deoxyuridine (EdU) staining

Proliferation of cells was investigated with BeyoClickTM EdU Cell Proliferation Kit (C0075S, Beyotime, China) according to the manufacturer's protocols. In brief, frozen brain sections were dried at temperature and permeated with 0.3% Triton X-100 in PBS for 30 min. Sections were then incubated with EdU working solution for 1 h at 37 °C in the dark. After incubation, regular immunofluorescence staining can be followed.

## Immunohistochemical staining

Frozen brain sections were dried at room temperature, and then pretreated with 0.3% $H_2O_2$ for 15 min to deactivate endogenous peroxidase. Sections were blocked with 3% normal sheep serum with 0.1% Tween 20 at room temperature for 2 h. Sections were then incubated in primary antibodies [rabbit anti-NeuN (1:500; Abcam, ab177487), rabbit anti-SOX2 (1:500; Millipore, ab5603), rabbit anti-TBR2 (1:500; Abcam, ab23345), mouse anti-PROX1 (1:200; Millipore, MAB5654), and rabbit anti-BLBP (1:500; Abcam, ab32423)] in blocking buffer overnight at 4 °C, followed by addition of the avidin-biotin-peroxidase complex (1:50; VECTASTAIN Elite ABC system, Vector Laboratories). Peroxidase was reacted in 3,3′-diaminobenzidine (5 mg/ml) and 0.075% $H_2O_2$ in Tris-HCl (pH 7.2). Sections were dehydrated in gradient ethanol (75% ethanol, 95% ethanol, 100% ethanol, and 100% ethanol, each for 5 min), and cleared twice in xylene for 5 min, then mounted in the neutral balsam.

## Behavior tests

We used 12- to 16-week-old age-matched male mice for all behavioral tests. Mice were housed (3–5 animals per cage) in standard filter-top cages with access to water and rodent chow at all times, maintained on a 12:12 h light/dark cycle (09:00–21:00 h lighting) at 22 °C, with relative humidity of 50–60%. All behavioral assays were done blind to

genotypes. Behavioral tests performed in current study include open field, rotarod, Morris water maze, fear conditioning, forced swimming, tail suspension, and elevated plus maze. Detailed procedures are described in Supplementary Methods.

## X-Gal staining

Frozen sections were fixed in fresh cold fixative (0.2% PFA) in buffer L0 (0.1 M PIPES buffer (pH 6.9), 2 mM $MgCl_2$, 5 mM EGTA) for 10 min. Slides were rinsed in PBS plus 2 mM $MgCl_2$ on ice, followed by a 10 min wash in the same solution. Place slides in detergent rinse [0.1 M PBS (pH 7.3), 2 mM $MgCl_2$, 0.01% sodium deoxycholate, 0.02% Nonidet P-40] on ice for 10 min. Slides were then moved to a freshly made and filtered X-Gal staining solution [0.1 M PBS (pH 7.3), 2 mM $MgCl_2$, 0.01% sodium deoxycholate, 0.02% Nonidet P-40, 5 mM $K_3Fe(CN)_6$, 5 mM $K_4Fe(CN)_6 \cdot 3H_2O$ and 1 mg/ml X-Gal]. Sections were incubated at 37 °C from a few minutes to overnight in the dark. Sections were rinsed with water to stop the reaction. Sections were dehydrated with gradient ethanol and xylene sequentially, and mounted with the neutral balsam.

## RNA isolation and reverse transcription (RT)

RNA isolation was performed using the RNAiso Plus (TAKARA) according to the manufacturer's instructions. Tissue or cells were homogenized using a glass-Teflon in 1 ml or 500 µl RNAiso Plus reagent on ice and phase separation was achieved with 200 µl or 100 µl chloroform. After centrifugation at 12,000 × $g$ for 15 min at 4 °C, RNA was precipitated by mixing aqueous phase with equal volumes of isopropyl alcohol and 0.5 µl 20 mg/ml glycogen. Precipitations were dissolved in DNase/RNase-free water (not diethylpyrocarbonate treated, Ambion). 1 µg of total RNA was converted to cDNA using M-MLV reverse transcriptase (TAKARA) under standard conditions with oligo(dT) or random hexamer primers and Recombinant RNase Inhibitor (RRI, TAKARA). Then the cDNA was subjected to quantitative RT-PCR (qRT-PCR) using the SYBR green assay with 2× SYBR green qPCR master mix (Bimake). The thermal profile was 95 °C for 5 min and 40 cycles of 95 °C for 15 s and 60 °C for 20 sec. *Gapdh* was used as endogenous control genes. The relative expression level for target genes was normalized by the Ct value of *Gapdh* using a $2^{-\Delta\Delta Ct}$ relative quantification method. Reactions were run on a CFX Connect TM Real-Time PCR Detection System (Bio-Rad). The primers used are listed in Supplementary Table 3.

## Neurosphere culture

Mouse hippocampal neural progenitor cells (NPCs) were enriched from P0 mouse hippocampi, cultured on ultra-low-attachment plates (Corning, New York, United States) and maintained in indicated culture media (DMEM/F12, Life Technologies) containing N2 and B27 supplements (1×, Life Technologies), 1 mM Na-pyruvate, 1 mM N-acetyl-L-cysteine (NAC), human recombinant FGF2, and EGF (20 ng/mL each; Life Technologies). After cultured in vitro for three generations, neurospheres were subjected to RNA-seq and ChIP-seq analyses.

## RNA-seq library construction

Total RNA was extracted as described above. The concentration and quality of RNA was measured with Nanodrop 2000c (Thermo Fisher Scientific) and an Agilent 2100 Bioanalyzer (Agilent Technologies), respectively. RNA-seq libraries were constructed by NEBNext® Ultra™ II RNA Library Prep Kit for Illumina® (NEB #E7775). Briefly, mRNA was extracted by poly-A selected with magnetic beads with poly-T and transformed into cDNA by first and second-strand synthesis. Newly synthesized cDNA was purified by AMPure XP beads (1:1) and eluted in 50 µl nucleotide-free water. RNA-seq libraries were sequenced by Illumina NovaSeq 6000 platform with pair-end reads of 150 bp. The sequencing depth was 60 million reads per library.

## Bulk RNA-seq data analysis

P0 hippocampus and neurosphere RNA-seq data were checked for quality control by FastQC (version 0.11.9). Paired-end reads were trimmed to remove adaptors and low-quality reads and bases using cutadapt (version 3.2). Clean reads were aligned to the mouse UCSC mm10 genome using STAR (version 2.7.10b) with default parameters. The number of covering reads were counted using featureCounts (version v2.0.1). The resulting read counts were processed with R package DESeq2 (version 1.38.1) to identify differential expression genes (log2 fold change > 0.4 and $p$ value < 0.05) between datasets. Cufflinks package (version 2.2.1) assembles individual transcripts from reads that have been aligned to reference genome. The gene expression level was normalized by fragments per kilobase of bin per million mapped reads (FPKM). Gene Ontology (GO) analysis and Gene Set Enrichment Analysis (GSEA) in this study were performed using clusterProfiler (version 4.2.2).

## Chromatin immunoprecipitation (ChIP) assay

For each experiment, single-cell suspensions from P0 hippocampi were collected as described above. The hippocampal tissue was digested into single cells by Papain (20U dissolved in each mL of DMEM/F12 medium, preheated at 37°). Cells were cross-linked with 1% formaldehyde for 10 min at room temperature, and quenched with 0.125 M of glycine for 5 min. Cross-linked samples were then rinsed twice in PBS, then cells were collected by centrifugation. Next, cells were pretreated with lysis buffer (50 mM of Tris-HCl [pH 8.0], 0.1% SDS, and 5 mM of EDTA) and incubated for 5 minutes with gentle rotation at 4 °C. After centrifugation, bottom cells were washed for two times with ice-cold PBS and harvested in ChIP digestion buffer (50 mM of Tris-HCl [pH 8.0], 1 mM of $CaCl_2$, and 0.2% Triton X-100). DNA was digested to 150-300 bp by micrococcal nuclease (NEB; M0247S). Sonicate cells in EP tubes with power output 100 W, 5 min, 0.5 s on, 0.5 s off on ice. The resulting lysate was centrifugation and divided into four parts for 10% input, H2AK119Ub, H3K27me3, and H3K36me2 immunoprecipitation. After diluting each sample (in addition to input) to 1 mL with dilution buffer (20 mM of Tris-HCl [pH 8.0], 150 mM of NaCl, 2 mM of EDTA, 1% Triton X-100, and 0.1% SDS), immunoprecipitation was further performed with sheared chromatin and 3 µg rabbit anti-H2AK119Ub antibody (CST, 8240 S); or rabbit anti-H3K27me3 antibody (CST, 9733S); or rabbit anti-H3K36me2 antibody (CST, 2901S), then incubated with protein A/G beads overnight at 4 °C on a rotating wheel. The next day, beads were wash with Wash Buffer I (20 mM Tris-HCl, pH 8.0; 1% Triton X-100; 2 mM EDTA; 150 mM NaCl; 0.1% SDS), Wash Buffer II (20 mM Tris-HCl, pH 8.0; 1% Triton X-100; 2 mM EDTA; 500 mM NaCl; 0.1% SDS), Wash Buffer III (10 mM Tris-HCl, pH 8.0; 1 mM EDTA; 0.25 M LiCl; 1% NP-40; 1% deoxycholate) and TE buffer. DNA was eluted by ChIP elution buffer (0.1 M of $NaHCO_3$, 1% SDS, 20 µg/mL of proteinase K). The elution was incubated at 65 °C overnight, and DNA was extracted with a DNA purification kit (DP214-03; TIANGEN).

## ChIP-seq library construction

ChIP-seq libraries were constructed by VAHTS Universal DNA Library Prep Kit for Illumina V3 (Vazyme ND607). Briefly, 50 µL purified ChIP DNA (5 ng) was end-repaired for dA tailing, followed by adaptor ligation. Each adaptor was marked with a barcode of 6 bp which can be recognized after mixing different samples together. Adaptor-ligated ChIP DNA was purified by VAHTS DNA Clean Beads (Vazyme N411) and then amplified by PCR of 10 cycles with primers matching with adaptors' universal part. Amplified ChIP DNA was purified again using VAHTS DNA Clean Beads in 35-µL EB elution buffer. For multiplexing, libraries with different barcode were mixed together with equal molar quantities by considering appropriate sequencing depth (about 30 million reads per library). Libraries were sequenced by Illumina Nova-seq 6000 platform with pair-end reads of 150 bp.

## ChIP-seq data analysis

DNA libraries were sequenced on Illumina NovaSeq 6000 platform. All P0 hippocampus and neurosphere ChIP-seq data were checked and removed with adaptor sequences same as the RNA-seq data processing. Clean reads were aligned to the mouse UCSC mm10 genome using Bowtie2 (version 2.4.5). Duplicates were removed using the samtools rmdup module. Regions of peaks were called using the SICER software package, with the input genomic DNA as a background control (parameters: -w 200 -rt 1 -f 150 -egf 0.77 -fdr 0.01 -g 600 -e 1000 --significant_reads). The bigwig signal files were visualized using the computeMatrix, plotHeatmap, plotProfile modules in Deeptools (version 3.5.1). Homer was used to identify adjacent genes from the peaks obtained from SICER.

## ATAC-seq library construction

ATAC-seq libraries were constructed by TruePrep DNA Library Prep Kit V2 for Illumina (Vazyme TD501). Briefly, P0 hippocampi were dissected and gently homogenized in cold nuclear isolation buffer (10 mM Tris-HCl, pH 7.4, 10 mM NaCl, 3 mM MgCl$_2$, 0.1% Igepal CA-630). Nuclei were collected by centrifugal precipitation. 50,000 nuclei were put into the tagmentation reaction for each sample (performed with 30 min incubation time at 37 °C). Immediately following the tagmentation, DNA fragments were purified using VAHTS DNA Clean Beads (2X). Purified DNA fragments were added with Illumina i5 + i7 adapters with unique index to individual samples followed by PCR reaction (PCR program: 72 °C for 3 min, 98 °C for 30 s, 98 °C for 15 s, 60 °C for 30 s, 72 °C for 30 s, repeat 3–5 for 13 cycles, 72 °C for 5 min, and hold at 4 °C). Generated libraries were purified using VAHTS DNA Clean Beads (1.2X). For multiplexing, libraries with different barcode were mixed together with equal molar quantities by considering appropriate sequencing depth (about 50 million reads per library). Libraries were sequenced by Illumina Nova-seq 6000 platform with pair-end reads of 150 bp.

## ATAC-seq data analysis

P0 hippocampus ATAC-seq raw data were trimmed by Cutadapt with parameters -u 3 -u −75 -U 3 -U −75 -m 30 and then aligned to mouse mm10 genome using Bowtie2 (-X 2000 --very-sensitive). Subsequently, we downloaded blacklisted regions including a large number of repeat elements in the genome from ENCODE project and then removed these significant background noise. ATAC-seq datasets contained a large percentage of reads that were derived from mitochondrial DNA. We removed mitochondrial reads after alignment using Samtools. Then, we filtered reads to remove exact copies of DNA fragments that arise during PCR using Picard's MarkDuplicates (version 2.26.4). All reads aligning to the + strand were offset by +4 bp, and all reads aligning to the − strand were offset −5 bp, since Tn5 transposase has been shown to bind as a dimer and insert two adaptors separated by 9 bp. We adjusted the shift read alignment using alignmentSieve. Next, peaks calling was finished by Macs2 (version 2.2.7.1) with parameters -f BAMPE --nomodel --keep-dup all --shift −100 --extsize 200 -g mm --cutoff-analysis -B. We created bigwig files for visualizing using bamCoverage (parameters: --normalizeUsing RPGC −effectiveGenomeSize 2407883318) in deepTools. Homer took narrow peak files as input and checked for the enrichment of both known sequence motifs and de novo motifs.

## Defining genomic features

Mm10 CpG island (CGI) regions were downloaded from the UCSC genome browser database. Promoters were defined as all mouse UCSC mm10 gene TSSs, extended by 2 kb upstream and downstream. CGIs in promoters were defined as ± 2 kb around CGI centers overlap with promoter regions. Overlapped regions between CGIs and promoters were identified using bedtools (version 2.29.2) intersect with parameters -e -f 0.5 -F 0.5. H2AK119ub1 +/− promoter genomic loci were defined as such promoter location with or without H2AK119ub1.

## In utero electroporation (IUE)

In utero microinjection and electroporation were performed as followed. Pregnant CD-1 mice with E14.5 embryos were anesthetized by injection of pentobarbital sodium (70 mg/kg), and the uteri were exposed through a 2 cm midline abdominal incision. Embryos were carefully pulled out using ring forceps through the incision and placed on sterile gauze wet with 0.9% sodium chloride. Plasmid DNA (prepared using Endo Free plasmid purification kit, Tiangen) mixed with 0.05% Fast Green (Sigma) was injected through the uterine wall into the telencephalic vesicle using pulled borosilicate needles (WPI). For gain-of-function experiments, pCIG (1 μg/μl) was mixed with pCAGGS-Wnt3a, pCAGGS-Wnt5a, pCAGGS-Wnt5b, pCAGGS-Wnt7b, pCAGGS-Wnt8b, (Wnt-mix) (0.5 μg/μl each), or with pCAGGS-SFRP2 (2 μg/μl). Control mice were injected with pCIG (1 μg/μl). Five electric pulses (33 V, 50 ms duration at 1 s intervals) were generated using CUY21VIVO-SQ (BEX) and delivered across the head of embryos using 5 mm forceps-like electrodes (BEX). The uteri were then carefully put back into the abdominal cavity, and both peritoneum and abdominal skin were sewed with surgical sutures. The whole procedure was completed within 30 min. Mice were warmed on a heating pad until they regained consciousness and were treated with analgesia (ibuprofen in drinking water) until sacrifice at E18.5.

## Plasmid construction

Full-length mouse *Wnt3a* and *Wnt8b* were amplified from cDNAs of E14.5 mouse hippocampi and then cloned into pCAGGS. Full-length *Wnt5a*, *Wnt5b*, *Wnt7b* were amplified from cDNAs of E16.5 mouse cortex, and then cloned into pCAGGS. Full-length mouse *Sfrp2* was amplified from cDNAs of P0 mouse cortex and then cloned into pCAGGS. The primers used are listed in Supplementary Table 4.

## Quantification and statistical analysis

Sections used for quantification were position-matched for control and experimental brains. Three corresponding brain cross sections of each control and cKO samples were selected for cell count statistics and the total number of these three cross sections was counted.

The length or area of adult hippocampal CA and DG were quantified by ImageJ. By dividing the average cell count (per section) by the average DG area (per section), the cellular density within the DG can be determined.

Images were binned against proximal-distal transverse axis to quantify the intensity of ISH or LacZ signals of hippocampi. A plot of normalized average signal intensity with standard error of the mean across those regions was generated using ImageJ.

Statistical tests were performed using GraphPad Prism (version 8.0.2). Data analyzed by unpaired two-tailed *t*-test were pre-tested for equal variance by *F*-tests. Unpaired Student's *t*-tests (two-tailed) were chosen when the data distributed with equal variance. For normally distributed data with unequal variance, an unpaired *t*-test with Welch's correction was used. One-way ANOVA followed by Tukey post hoc test was used for multiple group comparison. Significant difference is indicated by a *p* value < 0.05 (*$p < 0.05$, **$p < 0.01$, ***$p < 0.001$, ****$p < 0.0001$). No statistical methods were used to pre-determine sample sizes but our sample sizes are similar to those reported in previous publications. Experiments were not randomized. Investigators were blinded as to the animal genotype during tissue section staining, image acquisition, and image analysis.

## Reporting summary

Further information on research design is available in the Nature Portfolio Reporting Summary linked to this article.

## Data availability

The RNA-seq, ChIP-seq and ATAC-seq data generated in this study have been deposited in the NCBI GEO database under accession GSE222465. RNA-seq data of P0 hippocampus, P0 neurosphere and P0 cortex have been deposited at GEO: GSE222464. ChIP-seq and ATAC-seq data of P0 hippocampus and P0 neurosphere in this study have been deposited at GEO: GSE222463 and GSE222462. Source data are provided with this paper. Any additional information required to analyze the data in this paper is available from authors upon reasonable request. Source data are provided with this paper.

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

## Acknowledgements

We thank Dr. Junlei Chang for providing the BAT-Gal mice. We thank the Core Facility and the Animal Facility of the Medical Research Institute of Wuhan University for technical support. We thank all Zhou lab members for the critical reading of the manuscript. Y. Zhou was supported by grants from the National Key R&D Program of China (2022YFA0806603 and 2018YFA0800700), National Natural Science Foundation of China (31970770 and 32270876) and the Fundamental Research Funds for the Central Universities (2042022dx0003). Y.L. was supported by grants from the National Key R&D Program of China (2018YFA0800700 and 2022YFA0806603), the National Natural Science Foundation of China (31970676) and Hubei Natural Science Foundation (2022CFB128).

## Author contributions

All authors participated in the scientific discussion. Y. Zhou conceived the research. B.Z. and Y. Zhou designed the experiments. B.Z., W.S., W.L., Y. Zheng, X.K., and J.W. carried out the experimental studies and analyses. C.Z. carried out the computational studies. Y. Zhou and B.Z. wrote the manuscript. Y. Zhou and X.W. revised the manuscript. All authors commented on the manuscript. Y. Zhou, Y.L., and T.Z. supervised the project.

## Competing interests

The authors declare no competing interests.
