## [Peer Review File · Nature Communications]

KDM2B regulates hippocampal morphogenesis by transcriptionally silencing Wnt signaling in neural progenitorsReviewer #1 (Remarks to the Author):

In this study entitled "KDM2B regulates hippocampal morphogenesis by transcriptionally silencing Wnt signalling in neural progenitors" Zhang et al examine the consequences of the embryonic CNS specific expression of a mutant form of the chromatin associated protein KDM2B, where this association is prevented. KDM2B is a histone demethylase that can also mediate the Polycomb repressive complex 1 recruitment to the chromatin. The dentate gyrus appears prominently affected in mutants. Using complementary deletion approaches, the authors present a compelling demonstration of the role of KDM2B for DNE progenitor migration in neonates. Molecular analyses further suggest that de-repression of Wnt signalling pathway occurs in mutants. However, it is unclear which activity of KDM2B, affected in mutants, underlies the phenotype. This manuscript presents a thorough and well-executed analysis of both morphological/cellular DG development. The quantity of data presented is impressive and very interesting.

The manuscript should be improved by providing some details and data, as listed below.

Concerning generation of the mutant mouse line (Fig. S1A,B), it was not clear from the scheme what the size of the PCR fragment is after deletion and whether this was shown on the panel B (we have floxed and Cre positive samples but do we have floxed;Cre double positive?). It would be good to indicate in panel A where the genotyping primers are located and expected PCR fragment sizes. In Fig.S1C, please indicate percentages. In Fig. S1F,G, there is still staining in the mutants, especially in the thalamus. Is this related to Cre efficiency or activity pattern? In Fig.1H, please indicate and document size changes and meaning of arrows.

Regarding the analysis of the DG phenotype, the authors mention that there are fewer neurons and stem cells in mutants, however it is unclear whether this relates to the DG being smaller (and this reduction is not quantified either and should be) or if in the smaller DG these cell types are less represented than in the control. Moreover, numbers of HOPX+, PROX1, and DCX+ cells are not shown.

What is the expression pattern of KDM2B during DG development? Which cell type express it and when? This could help defining its role better.

There appears to be an ectopic formation of a scaffold in the DNE region, along the secondary matrix (Fig.3A-B) which could explain why more progenitors are generated and can delaminate and migrate but are unable to reach their destination, unsupported as this abnormal scaffold does not expand to the forming DG. The author did not quantify or examine these supporting GFAP cells (difficult to see on the composite panels) which normally exclusively form in the fimbria. It would be interesting to investigate, or at the very least discuss, this aspect further.

Regarding the CA phenotype, a reduction of calbindin positive cells is shown in Fig.1. Later, in Fig.6 the effect of Wnt genes and Sfrp2 overexpression are examined by IUE in this domain. The CA phenotype could be further explored while, in contrast, effect of the electroporation should have been investigated in the DG region.

Regarding the behavioural characterisation of the cKO animals, the moving tracks from the open field test (Fig. 2M) show very different patterns in the cKO and significant reduction in their distance and mobility time, while the quantification shows no differences (Fig. S3.A), please explain this apparent discrepancy.

While reporter analyses indicate that Nestin-CreERT2 (Fig. S3) is active on the reporter allele, this does not necessarily mean that deletion of the Kdm2b exon was as efficiently induced. Striking differences in deletion efficiency exist between different alleles, and this is particularly true for inducible Cre which are only active during tamoxifen treatment. Therefore, unless deletion of the targeted allele is shown, these experiments are suggestive but not conclusive.

Regarding the molecular analysis of cKO DG development It is unclear from the text why the

genes in Fig. 6.D-G were picked for analysis. Probably because these are Wnt pathways genes mostly enriched in CpG islands? If so, authors should mention it in the text.

Fig.5I, L, O, are the differences observed significant?

From the ATAC profiles in Fig. 6.H and S8, chromatin do not seem particularly more open in the loci shown. Authors should comment on this, and improve representations, the figures are quite difficult to read (lots of data but very difficult to see differences between mutants and controls). Is Lef-1 profile exclusively affected in neurospheres (Fig.S8K)?

Although not pivotal for the main scope of the paper, data shown in Fig. S9.A-G is not particularly straightforward. The line of the side of panel A ("High in Ctx") appears too short and should go up to Wnt6. The interpretation provided, regarding the differential effect of Wnt signalling dysregulation by KDM2B in the cortex vs. the hippocampus, is not supported by the data presented. Many Wnt genes are indeed clearly upregulated in the cortex. Are there statistically more Wnt pathway DEGs in DG compared to the cortex?

Regarding the generation of Rnf2 mouse line for analysis of Ring1B role in DG development, comes a bit unexpected, while being useful for exploring the specific role of KDM2B in DG development. Authors should mention Ring1B and its function in the intro already to facilitate the inclusion of this chapter. In addition, histone modifications shown in Fig. S9.I should be quantified.

Finally, given the unexpected results from these Rnf2 mutants, the potential role of KDM2B for DG development should be better discussed. The graphical abstract suggests a potential role for PRC1 while the Rnf2 mutant phenotype may suggest otherwise?

Minor comments:

- Line 68: define CBX proteins.
- Line 68,69: should put "canonical PRC1" and "variant PRC1" in brackets to improve readability.
- Fig. 5 should correct to "upregulated" and "biological process".
- Line 134-141: S1C is referred but should be replaced by S2C.
- Line 150: the authors indicate data not shown, but it seems to refer to Fig.3E?
- Cartoons used in Fig.3.B,D,J; 4.B; S4.B,D; S5.B; S9.J are strikingly similar to Caramello et al., 2021 and should therefore be referred to as "adapted from".
- In line 280, the length of the EdU pulse for the analysis at E16.5 should be mentioned.
- In Fig. S6 the title should be modified as "abnormal/precocious neuronal differentiation is not responsible...."
- "a tad", "climax", "block of their attenuation" should be replaced.
- Please define all abbreviations

Reviewer #2 (Remarks to the Author):

In this manuscript, Zhang, et al have discovered that KDM2B plays a crucial role in regulating hippocampal morphogenesis by silencing Wnt signaling in neural progenitors. When the chromatin-association capability of KDM2B was removed in developing dorsal telencephalon, the hippocampus became drastically smaller with disorganized cellular components and structure. KDM2B mutations in mice resulted in defects in spatial memory, motor learning, and fear conditioning. The migration and differentiation of neural progenitor cells were greatly impeded in the developing hippocampus. The study suggests that KDM2B is essential for proper hippocampal formation and function. The manuscript presents several concrete advances in understanding the role of KDM2B in hippocampal morphogenesis. Overall, I think this is valuable manuscript.

However, there are a few points that could be considered as potential limitations:

1. The behavioral tests used in the study, such as the Morris water maze and fear conditioning, provide valuable insights into the functional consequences of KDM2B mutations. However, these tests have their own limitations and may not fully capture the complexity of behaviors or cognitive functions that could be affected by KDM2B mutations.
2. The study found that removal of KDM2B's chromatin association capability in adult neural stem cells exerts no effect on adult neurogenesis of the dentate gyrus. This could be seen as a limitation as it suggests that the role of KDM2B in neurogenesis may be limited to the developmental stage.

Reviewer #3 (Remarks to the Author):

In the manuscript by Zhang et al. authors evaluated the development of the hippocampus when removing the chromatin association capability of KDM2B (Kdm2b Δ CxxC, henceforth Δ CxxC) in progenitor cells of the developing dorsal telencephalon. They found an abnormal morphology of the hippocampus in adult Δ CxxC conditional KO (cKO) mice, particularly in the dentate gyrus which was smaller. Δ CxxC mice also showed cognitive and motor impairments. The proposed mechanism involves the increase in the expression of several components of the Wnt signaling pathway that were de-repressed. These are very interesting and novel findings. Most of the conclusions are well supported by the results obtained.

Major comments:

1. A description of the expression patterns of Emx1 and Nestin must be included, as well as a discussion of the differences observed in the survival of both conditional KO (cKO) mice. What is the phenotype of Nestin- Δ CxxC (compared to Emx1- Δ CxxC) at P0 and P7?
2. The same analysis shown in Fig. S1H should be carried out in Emx1- Δ CxxC
3. Authors determined that a strong decrease in GFAP+SOX2+ NSCs, HopX+ NSCs, TBR2+ and DCX+ (which should be quantified as done for NeuN+ and GFAP+SOX2+) cells in the dentate gyrus of adult Δ CxxC animals. However, they conclude that deletion of CxxC ZF of KDM2B had no effect on adult neurogenesis of the DG based on the conditional deletion in adult animals. Authors should improve the discussion of these results.
4. The density (per GCL area or volume) of GFAP+SOX2+ NSCs, HopX+ NSCs, TBR2+ and DCX+ cells should be quantified to demonstrate whether the decrease of these cells is due to the shorter DG.
5. Authors conclude that Δ CxxC mice exhibit hippocampal agenesis. I suggest changing "agenesis" since although smaller, the hippocampus is present in cKO mice.
6. Figure 2F is not mentioned in the Results section or discussed. What can be concluded of this analysis where no differences in the number of platform crossing were observed in cKO mice. The representative swim path of cKO mice during the probe trial (Fig 2C) should be change; it does not seem representative based on the graph.
7. Line 235-237: "Together, loss of KDM2B-CxxC impedes migration and differentiation of IPCs, hence proper production and localization of granule neurons during hippocampal formation." How did authors conclude with the results of Figure 3 that there was an impairment in differentiation. The effects in PROX1+ cells could be caused only by the altered migration; and is it possible to exclude a proliferation impairment? Total number of EdU+ cells and the percentage of EdU+ cells positive for PROX1 should be evaluated.
8. Line 288-289: "In summary, the migratory and differentiating trajectory of hippocampal progenitors were greatly delayed upon loss of KDM2B-CxxC". What do authors mean by delayed differentiating trajectory? Considering that as shown in Fig 4 "the differentiating rate from RGCs to IPCs (PAX6+TBR2+/PAX6+) at FDJ is 60% higher in cKOs".

9. Line 298-301: "Together, KDM2B regulates hippocampal morphogenesis by controlling multiple behaviors, including coordinated RGC to IPC differentiation, migration, and divisions of neural progenitors (Fig. S6C)." What is the evidence to conclude that KDM2B regulates divisions of neural progenitors? The discussion of the evidence supporting the specific effects (or roles of KDM2B) on migration, differentiation, proliferation should be improved.

10. Please indicate in the results section what Wnt ligands were used in the "Wnt-mix".

Minor comments:

- Line 139, should say Fig S2C instead of S1C.
- Line 164, should say Fig 2D-2E instead of 2C-2F.
- In the description of the Morris water maze results, it should be indicated that the probe trial was carried out on day 6.
- There is no sequential order in the description of some Figures. For example, Fig 2G is described before Fig 2D, and Fig S8 before Fig S7. This occurs in the description of several Figures and must be modified.
- Figure S3 legend, line 75 "(D)" not "(A)".
- Fig3I, y-axis "Total" not "Totol"
- A brief description of the markers used should be given in the results section when describing the IFs (e.g. ZBTB20).

REVIEWER COMMENTS

Reviewer #1 (Remarks to the Author):

In this study entitled “KDM2B regulates hippocampal morphogenesis by transcriptionally silencing Wnt signalling in neural progenitors” Zhang et al examine the consequences of the embryonic CNS specific expression of a mutant form of the chromatin associated protein KDM2B, where this association is prevented. KDM2B is a histone demethylase that can also mediate the Polycomb repressive complex 1 recruitment to the chromatin. The dentate gyrus appears prominently affected in mutants. Using complementary deletion approaches, the authors present a compelling demonstration of the role of KDM2B for DNE progenitor migration in neonates. Molecular analyses further suggest that de-repression of Wnt signalling pathway occurs in mutants. However, it is unclear which activity of KDM2B, affected in mutants, underlies the phenotype. This manuscript presents a thorough and well-executed analysis of both morphological/cellular DG development. The quantity of data presented is impressive and very interesting.

The manuscript should be improved by providing some details and data, as listed below.

We are encouraged by the reviewer’s overall positive opinion and thank him/her for all very constructive suggestions. We’ve performed additional experiments and analyses, and have extensively revised the manuscript accordingly. Concerning “which activity of KDM2B, affected in mutants, underlies the phenotype”, we believe we’ve presented sufficient evidence showing KDM2B facilitates proper silencing of Wnt signaling during hippocampal morphogenesis, which is associated with KDM2B’s functions in targeting the variant polycomb repressive complex 1 to the chromatin to mediate repressive histone modifications. Below is our point-by-point response.

1. Concerning generation of the mutant mouse line (Fig. S1A,B), it was not clear from the scheme what the size of the PCR fragment is after deletion and whether this was shown on the panel B (we have floxed and Cre positive samples but do we have floxed;Cre double positive?). It would be good to indicate in panel A where the genotyping primers are located and expected PCR fragment sizes. In Fig.S1C, please indicate percentages. In Fig. S1F,G, there is still staining in the mutants, especially in the thalamus. Is this related to Cre efficiency or activity pattern? In Fig.1H, please indicate and document size changes and meaning of arrows.

We appreciate these suggestions. In revised Supplementary Fig.1a, locations of genotyping primers were indicated. We used toe tissues for genotyping, which is not edited by the *Emx1*-Cre. In revised Supplementary Fig.1b, genotypes of each animal were labelled on the top. In revised Supplementary Fig.1c, percentages of each genotype were also shown. *Emx1*-Cre is specifically active in the dorsal forebrain (including developing neocortex and hippocampus), but not most ventral structures such as the thalamus (revised Supplementary Fig.1h). Therefore, the absence of the *Kdm2b in situ* hybridization (ISH) signal in the cKO neocortex and hippocampi but not in the thalamus reflects the spatiotemporal activity of *Emx1*-Cre. In revised Supplementary Fig.1k and 1l, black arrows indicated wild-type KDM2B, whereas red arrows indicated CxxC-ZF deleted KDM2B, and the size reductions (8.3 kDa) were documented in the manuscript.

2. Regarding the analysis of the DG phenotype, the authors mention that there are fewer neurons and stem cells in mutants, however it is unclear whether this relates to the DG being smaller (and this reduction is not quantified either and should be) or if in the smaller DG these cell types are less represented than in the control. Moreover, numbers of HOPX+, PROX1, and DCX+ cells are not shown.

We thank the reviewer to point this out and performed additional staining and

measurements. In revision, we measured the length and area of DGs to find *Kdm2b*^{cKO} hippocampi had significant smaller DGs (Supplementary Fig.2d-s2g). The decrease of numbers of NeuN+ neurons in cKO animal was comparable to the decrease of DG length or area. Interestingly, the cell densities of HopX+ and SOX2+GFAP+ stem cells were significantly decreased upon deletion of KDM2B-CxxC (Supplementary Fig.2m-2n). Moreover, numbers of HopX+, PROX1+, and DCX+ cells were shown in revision (Fig.1i, Supplementary Fig.2i-2j).

3. What is the expression pattern of KDM2B during DG development? Which cell type express it and when? This could help defining its role better.

This is also a very important issue. We first performed qRT-PCR of *Kdm2b* using developing hippocampal tissues. The expression of *Kdm2b* showed an expression plateau between E16 and P0, when hippocampal morphogenesis and neurogenesis are at the peak (Supplementary Fig.1i). Then *Kdm2b* ISH were performed onto P0 hippocampal sections followed by TBR2 immunofluorescent staining. Data showed many *Kdm2b*-expressing DG cells also express TBR2, strongly suggesting *Kdm2b*'s role in these cells (Supplementary Fig.1j). Related texts were modified or added accordingly.

4. There appears to be an ectopic formation of a scaffold in the DNE region, along the secondary matrix (Fig.3A-B) which could explain why more progenitors are generated and can delaminate and migrate but are unable to reach their destination, unsupported as this abnormal scaffold does not expand to the forming DG. The author did not quantify or examine these supporting GFAP cells (difficult to see on the composite panels) which normally exclusively form in the fimbria. It would be interesting to investigate, or at the very least discuss, this aspect further.

We thank the suggestion. The ectopic formation of scaffolds in the DNE region along the secondary matrix (2ry) could be one of reasons that

progenitors could not reach their destination. Per suggestion by the reviewer, we did quantify the GFAP density at the border of fimbria. Data showed that the GFAP density is not altered in the 2ry, but slightly increased in the fimbria, probably due to the compression by amassed TBR2+ cells at the 2ry (Fig.3o). We've stated the point in revision. Moreover, bearing the possibility in mind, we did use *Nestin-Cre* to ablate *Kdm2b-CxxC* in original submission. As we've tested, *Nestin-Cre* is mostly not active in cortical hem (CH) derived astrocytic scaffolds at the fimbria (Supplementary Fig.5b). In the *Kdm2b^{Nestin-ΔCxxC}* hippocampi, significantly more IPCs still accumulated at the DNe and the FDJ, but fewer IPCs at the DG, with the fimbria compressed (Supplementary Fig.5c-5g).

5. Regarding the CA phenotype, a reduction of calbindin positive cells is shown in Fig.1. Later, in Fig.6 the effect of Wnt genes and *Sfrp2* overexpression are examined by IUE in this domain. The CA phenotype could be further explored while, in contrast, effect of the electroporation should have been investigated in the DG region.

Regarding these two points, we further analyzed phenotypes in cKO CA and electroporated DG. First, EdU was administered at E14.5 followed by phenotypic analyses at E18.5. In cKO CA1 region, significantly more EdU+ cells resided at VZ and fewer EdU+ cells located at pyramidal cell layer. Moreover, significantly more EdU-labelled cells expressed TBR2. Second, at E16.5 cKOs, more PAX6+ and TBR2+ cells were detected at the hippocampal neuroepithelia (HNE), where CA neurogenesis occurs. While more PAX6+ cells were proliferative (PAX6+EdU+), a smaller portion of TBR2+ cells were dividing (Supplementary Fig.7). Thus, loss of KDM2B-CxxC also leads to hampered neurogenesis at the CA region. Third, the effect of Wnt-mix and SFRP2 electroporation on the DG was analyzed in revision. Significantly fewer electroporated GFP+ cells reached the distal (3ry) region of DG, while

more electroporated GFP+ cells expressed TBR2 at DG, indicating hampered neurogenesis upon overexpression of Wnt ligands and SFRP2 (Fig.7f, 7h, 7j).

6. Regarding the behavioural characterisation of the cKO animals, the moving tracks from the open field test (Fig. 2M) show very different patterns in the cKO and significant reduction in their distance and mobility time, while the quantification shows no differences (Fig. S3.A), please explain this apparent discrepancy.

We apologize for not explaining the point more clearly. The cKO animals displayed two major moving patterns in the open field test. Three out of nine were not willing to explore at all (Fig.2m, top-right), whereas four out of the rest six animals would circle around the wall but not explore the center area (Fig.2m, bottom-right). Tracks were shown in **Response fig.1**. Therefore, statistic analyses did not reveal significant difference between control and cKOs regarding distance, velocity and mobility time.

Response fig.1. Moving tracks of control (top) and *Kdm2b^{Emx1-ΔCxxC}* (bottom) mice.

7. While reporter analyses indicate that Nestin-CreERT2 (Fig. S3) is active on the reporter allele, this does not necessarily mean that deletion of the *Kdm2b*

exon was as efficiently induced. Striking differences in deletion efficiency exist between different alleles, and this is particularly true for inducible Cre which are only active during tamoxifen treatment. Therefore, unless deletion of the targeted allele is shown, these experiments are suggestive but not conclusive.

We agree with the reviewer that "*deletion of the Kdm2b exon might not be efficiently induced*" because we did not provide direct evidence of exon deletion in the adult DG. We'd like to point out that we've examined **FOUR** *Nestin-CreERT2;Ai14* mice (2 each for short-term and long-term chasing), and all showed efficient recombination/reporter labeling in both the SVZ and SGZ (**Response fig.2**). As the recombination efficiency of both *Nestin-CreERT2* and *Kdm2b^{flox(CxxC)}* mice were tested, it's very likely that deletion of the CxxC exon in *Kdm2b^{Nestin-CreERT2-ΔCxxC}* is efficient on tamoxifen treatment. Nonetheless, in revision, we've added a sentence in this part: 'We would like to point out that the possibility of inefficient deletion of Kd2mb-CxxC on TAM

treatment might exist.

Response fig.2. *Nestin-CreERT2;Ai14* mice were treated with DMSO or TAM. Brains were subjected to sagittal sectioning and DAPI staining. SVZ and SGZ regions were enlarged on the bottom.

8. Regarding the molecular analysis of cKO DG development It is unclear from the text why the genes in Fig. 6.D-G were picked for analysis. Probably because these are Wnt pathway genes mostly enriched in CpG islands? If so, authors should mention it in the text.

This is a very interesting point. First, The RNA-seq analysis indicated that many Wnt signaling genes, including those encoding Wnt ligands and SFRP2, were significantly upregulated in *Kdm2b^{Emx1-ΔCxxC}* cKO hippocampi, which

were validated by qRT-PCR (Fig.6d-6g, Supplementary Fig.10a-10f). Because it is known that KDM2B mediates gene silencing by targeting the variant polycomb repressive complex 1 (PRC1) to non-methylated CpG islands (CGIs), we then paid special attention to the change of H2AK119Ub (deposited by PRC1) and H3K27me3 on these de-repressed genes. Data showed CGIs of many de-repressed genes indeed lost the enrichment of H2AK119Ub and H3K27me3. Second, CGIs are largely localized at promoter regions. No gene preference has been reported regarding distributions of CGIs. The change of repressive marks on Wnt signaling genes is probably due to selective targeting by KDM2B. We've improved related descriptions in revision.

9. Fig.5l, L, O, are the differences observed significant?

We did statistical analysis for fig 5i, l, and o in revision and added values in revision.

10. From the ATAC profiles in Fig. 6.H and S8, chromatin do not seem particularly more open in the loci shown. Authors should comment on this, and improve representations, the figures are quite difficult to read (lots of data but very difficult to see differences between mutants and controls). Is Lef-1 profile exclusively affected in neurospheres (Fig.S8K)?

We apologize for the confusion. I like to emphasize the CGI regions for *Wnt7b*, *Wnt3*, *Sfrp2*, *Eomes* and *Neurod1* were more open in cKO hippocampi, whereas other regions were less obvious (Fig. 6h; Supplementary Fig. 10i). We've improved description on this in revision. The decrease of enrichment of H2AK119Ub and H3K27me3 at *Lef1*'s CGI was present but less obvious in cKO HP tissues (**Response fig.3**). We've

indicated the point in revision.

Response fig.3. The UCSC genome browser view of HA2K119ub, H3K27me3 and H3K36me2 enrichment in P0 control and *Kdm2b^{Emx1-ΔCxxC}* (cKO) neurosphere (top) and hippocampi (bottom) at the CGI promoter of the *Lef1* gene.

11. Although not pivotal for the main scope of the paper, data shown in Fig. S9.A-G is not particularly straightforward. The line of the side of panel A (“High in Ctx”) appears too short and should go up to *Wnt6*. The interpretation provided, regarding the differential effect of Wnt signalling dysregulation by KDM2B in the cortex vs. the hippocampus, is not supported by the data presented. Many Wnt genes are indeed clearly upregulated in the cortex. Are there statistically more Wnt pathway DEGs in DG compared to the cortex? We thank the reviewer for pointing this out. The line marking “high in Ctx” has been moved up to *Wnt6* in revised Supplementary Fig.11a. Regarding whether there are statistically more Wnt pathway DEGs in DG than those in the cortex, we did analyze expression levels of all Wnt ligand and receptor

genes in hippocampi (HP) and cortex (Ctx). Data showed these genes expressed at a higher level in HP than in Ctx in both control and cKO brains. Moreover, deletion of KDM2B-CxxC elevated their expressions in both HP and Ctx (**Response fig.4**). Numbers of up-regulated Wnt pathway genes were 59 in the HP versus 42 in the Ctx [data from RNA-seq of HP and Ctx (log₂ fold change > 0.4 and *P* value < 0.05)].

Response fig.4. The heat map (A) and box plots (B-C) showing expression of Wnt ligand and receptor DEGs in cKO hippocampus (HP) and the cortex (Ctx).

12. Regarding the generation of Rnf2 mouse line for analysis of Ring1B role in DG development, comes a bit unexpected, while being useful for exploring the specific role of KDM2B in DG development. Authors should mention Ring1B and its function in the intro already to facilitate the inclusion of this chapter. In addition, histone modifications shown in Fig. S9.I should be quantified.

We thank these suggestions and have mentioned Ring1B and its function in the introduction. In addition, quantification of immunoblotting of histone modifications were done in revision (Supplementary Fig.11i).

13. Finally, given the unexpected results from these Rnf2 mutants, the potential role of KDM2B for DG development should be better discussed. The graphical abstract suggests a potential role for PRC1 while the Rnf2 mutant phenotype may suggest otherwise?

This is an interesting and important point. As the enzymatic component of PRC1, Ring1B is likely responsible for genome-wide deposition of H2AK119Ub. In contrast, KDM2B could selectively control expressions of Wnt pathway genes by targeting variant PRC1 to their CGIs. We've discussed the point in initial submission: "KDM2B likely controls fate determination of hippocampal progenitors by selectively repressing a series of progenitor genes including those in the Wnt pathway via PRC1.1". We further added: "expression alterations of other targets of Ring1B and PRC1 might counter some effects caused by loss of KDM2B-CxxC" and "How KDM2B selectively targets these genes during hippocampal morphogenesis remains to be investigated" in revision.

Minor comments:

We apologize for all these imprecise expressions and have fixed them in revision accordingly with only one exception.

- Line 68: define CBX proteins.
- Line 68,69: should put "canonical PRC1" and "variant PRC1" in brackets to improve readability.
- Fig. 5 should correct to "upregulated" and "biological process".
- Line 134-141: S1C is referred but should be replaced by S2C.
- Line 150: the authors indicate data not shown, but it seems to refer to Fig.3E?

The quantification of PAX6+ and TBR2+ in E16.5 cKO neocortices was indeed not shown because it was not quite relevant.

- Cartoons used in Fig.3.B,D,J; 4.B; S4.B,D; S5.B; S9.J are strikingly similar

to Caramello et al., 2021 and should therefore be referred to as “adapted from”.

- In line 280, the length of the EdU pulse for the analysis at E16.5 should be mentioned.

- In Fig. S6 the title should be modified as “abnormal/precocious neuronal differentiation is not responsible.... “

- “a tad”, “climax”, “block of their attenuation” should be replaced.

- Please define all abbreviations

The abbreviations list is summarized in Supplementary Table 1.

Reviewer #2 (Remarks to the Author):

In this manuscript, Zhang, et al have discovered that KDM2B plays a crucial role in regulating hippocampal morphogenesis by silencing Wnt signaling in neural progenitors. When the chromatin-association capability of KDM2B was removed in developing dorsal telencephalon, the hippocampus became drastically smaller with disorganized cellular components and structure. KDM2B mutations in mice resulted in defects in spatial memory, motor learning, and fear conditioning. The migration and differentiation of neural progenitor cells were greatly impeded in the developing hippocampus. The study suggests that KDM2B is essential for proper hippocampal formation and function. The manuscript presents several concrete advances in understanding the role of KDM2B in hippocampal morphogenesis. Overall, I think this is valuable manuscript.

However, there are a few points that could be considered as potential limitations:

1. The behavioral tests used in the study, such as the Morris water maze and fear conditioning, provide valuable insights into the functional consequences

of KDM2B mutations. However, these tests have their own limitations and may not fully capture the complexity of behaviors or cognitive functions that could be affected by KDM2B mutations.

We thank the reviewer for raising the important issue. We also realized that although these behavioral tests were commonly used in studies related to hippocampal functions, they have their own limitations and may not fully capture the complexity of behaviors and/or cognitive functions that could be affected by KDM2B mutations. Moreover, as we mentioned in response to **Reviewer 1**, most *Kdm2b-CxxC* ckO mice were not willing to explore novel environment. Consistently, in novel object recognition and three-chamber social interaction tests, most ckO animals would stay in corners or circle around walls but were not willing to explore. Therefore, we only included data of non-autonomous behavior tests such as the Morris water maze, rotarod, and fear conditioning.

We have discussed these limitations in revision in the last paragraph of Discussion: “More specific behavioral tests, such as novel object recognition and two-choice spatial discrimination test, could better describe how hippocampal functions were affected by KDM2B mutations”.

2. The study found that removal of KDM2B's chromatin association capability in adult neural stem cells exerts no effect on adult neurogenesis of the dentate gyrus. This could be seen as a limitation as it suggests that the role of KDM2B in neurogenesis may be limited to the developmental stage.

According to data revealed in the study, we agree with the reviewer that the role of KDM2B is largely in hippocampal morphogenesis at prenatal and postnatal stages but not in adult neurogenesis of the dentate gyrus. We reasoned that KDM2B is dispensable for adult neurogenesis of DG is because KDM2B is expressed at a relatively lower level in adult hippocampus (Supplementary Fig.1i). We've discussed the point in revision: “Interestingly, deletion of KDM2B-CxxC in adult neural stem cells exerts no effect on DG

neurogenesis, probably reflecting transient expression of KDM2B in embryonic and postnatal hippocampi".

Reviewer #3 (Remarks to the Author):

In the manuscript by Zhang et al. authors evaluated the development of the hippocampus when removing the chromatin association capability of KDM2B (Kdm2b Δ CxxC, henceforth Δ CxxC) in progenitor cells of the developing dorsal telencephalon. They found an abnormal morphology of the hippocampus in adult Δ CxxC conditional KO (cKO) mice, particularly in the dentate gyrus which was smaller. Δ CxxC mice also showed cognitive and motor impairments. The proposed mechanism involves the increase in the expression of several components of the Wnt signaling pathway that were downregulated. These are very interesting and novel findings. Most of the conclusions are well supported by the results obtained.

Major comments:

1. A description of the expression patterns of *Emx1* and *Nestin* must be included, as well as a discussion of the differences observed in the survival of both conditional KO (cKO) mice. What is the phenotype of *Nestin*- Δ CxxC (compared to *Emx1*- Δ CxxC) at P0 and P7?

We appreciate these suggestions and have described the expression pattern of *Emx1*-Cre and *Nestin*-Cre in revision. In particular, reporter mice (*Emx1*-Cre;*Ai14*) confirmed the activity of *Emx1*-Cre in neural stem cells (radial glial cells) and their progeny of dorsal forebrain structures including the hippocampus (Supplementary Fig.1h). *Nestin*-Cre mice used in this study is from the Jackson Laboratories (stock number 003771) and is active in all neural stem cells from E10.5 (Supplementary Fig.5b). *Nestin*-Cre- Δ CxxC pups were born at a ratio lower than the Mendelian expectation and could not be obtained at P7, indicating non-neural effects on loss of KDM2B-CxxC (Response Table 1). The phenotype of *Nestin*-Cre- Δ CxxC at P0 has been

described in Supplementary Fig.5c-5g, which is similar to that of P0 *Emx1-Cre*-deltaCxxC, showing prominent hippocampal hypoplasia.

Stage	Total	Kdm2b ^{fl/fl or fl/+}	Kdm2b ^{Nestin-cre;fl/+}	Kdm2b ^{Nestin-cre;fl/fl}
Embryo				
E14.5	10	4 (40%)	2 (20%)	4 (40%)
E15.5	27	11 (40.7%)	11 (40.7%)	5 (18.5%)
E16.5	19	12 (63.2%)	1 (5.3%)	6 (31.6%)
Total	56	27 (48.2%)	14 (25%)	15 (26.8%)
Postnatal				
P0	63	35 (55.6%)	19 (30.2%)	9 (14.3%)
P7	13	8 (61.5%)	5 (38.5%)	0 (0%)

Response Table 1

2. The same analysis shown in Fig. S1H should be carried out in *Emx1*-deltaCxxC

We thank the reviewer for the suggestion and performed immunoblotting for P0 cortex of *Emx1-Cre-DeltaCxxC*, which showed almost the same pattern of truncations as found in *Nestin-Cre-DeltaCxxC* (Supplementary Fig.1k-1l).

3. Authors determined that a strong decrease in GFAP+SOX2+ NSCs, HopX+ NSCs, TBR2+ and DCX+ (which should be quantified as done for NeuN+ and GFAP+SOX2+) cells in the dentate gyrus of adult deltaCxxC animals.

However, they conclude that deletion of CxxC ZF of KDM2B had no effect on adult neurogenesis of the DG based on the conditional deletion in adult animals. Authors should improve the discussion of these results.

First, we've performed quantifications on the numbers of these cells (Fig. 1i, Supplementary Fig. 2i-2k). Second, data from *Kdm2b*^{Nestin-CreERT2-ΔCxxC} cKO mice did show the deletion of KDM2B-CxxC in adult DGs had no effect on the neurogenesis of the SGZ. In other words, the effects of KDM2B on hippocampus are at embryonic and postnatal stage but not at adult stage. The decrease of adult NSCs in the *Kdm2b*^{Emx1-ΔCxxC} SGZ is largely due to

hippocampal hypoplasia. We reasoned that it is because KDM2B is transiently expressed in embryonic and postnatal hippocampus (Supplementary Fig.1i). We've discussed the point in revision.

4. The density (per GCL area or volume) of GFAP+SOX2+ NSCs, HopX+ NSCs, TBR2+ and DCX+ cells should be quantified to demonstrate whether the decrease of these cells is due to the shorter DG.

In revision, we've quantified cell densities of GFAP+SOX2+ NSCs, HopX+ NSCs, TBR2+ and DCX+ cells, revealing that these cells were less dense in cKO DGs (Supplementary Fig. 2l-2p).

5. Authors conclude that deltaCxxC mice exhibit hippocampal agenesis. I suggest changing "agenesis" since although smaller, the hippocampus is present in cKO mice.

We apologize for the inaccuracy. In revision, "agenesis" is replaced with "hypoplasia" throughout the manuscript.

6. Figure 2F is not mentioned in the Results section or discussed. What can be concluded of this analysis where no differences in the number of platform crossing were observed in cKO mice. The representative swim path of cKO mice during the probe trial (Fig 2C) should be change; it does not seem representative based on the graph.

We apologize for not mentioning the panel and agree that there was no statistic difference regarding the number of platform crossing in cKO mice. We therefore replaced representative swim path of the cKO mice and mentioned it in revision (Fig. 2c). Nonetheless, cKO mice spent less time in the SE quadrant where the platform located (Fig. 2g), indicating that the cKO mice had impaired spatial memory.

7. Line 235-237: "Together, loss of KDM2B-CxxC impedes migration and

differentiation of IPCs, hence proper production and localization of granule neurons during hippocampal formation.” How did authors conclude with the results of Figure 3 that there was an impairment in differentiation. The effects in PROX1+ cells could be caused only by the altered migration; and is it possible to exclude a proliferation impairment? Total number of EdU+ cells and the percentage of EdU+ cells positive for PROX1 should be evaluated. We agree the reviewer that based on data presented in Figure 3, the decrease of PROX1+ DG neurons could be largely explained by impeded migration of TBR2+ progenitors. The sentence was then changed into “loss of KDM2B-CxxC impedes migration of IPCs, subsequently resulting in hampered production and localization of granule neurons during hippocampal formation” in revision. We’d like to point out that the number of PROX1+EdU+ cells was also reduced by 27.1% in cKO hippocampi (Fig. 3d), which suggests impaired neuronal production. We also quantified EdU+ cells and the percentage of EdU+ cells positive for PROX1 in the DG. Data showed 39.5% fewer EdU+ cells were detected in cKO DGs (Fig. 3e), whereas the ratio of EdU+PROX1+/PROX1+ was not significantly altered. We’d like to point out since it’s the birthdating (7 days pulse-chase) experiments, we could not infer that in cKO hippocampi, cells born at E15.5 (labelled by EdU) had fewer chances to become granule cells and located at the DG. The issue of proliferation will be addressed in the following section.

8. Line 288-289: “In summary, the migratory and differentiating trajectory of hippocampal progenitors were greatly delayed upon loss of KDM2B-CxxC”. What do authors mean by delayed differentiating trajectory? Considering that as shown in Fig 4 “the differentiating rate from RGCs to IPCs (PAX6+TBR2+/PAX6+) at FDJ is 60% higher in cKOs”.

9. Line 298-301: “Together, KDM2B regulates hippocampal morphogenesis by controlling multiple behaviors, including coordinated RGC to IPC differentiation, migration, and divisions of neural progenitors (Fig. S6C).” What

is the evidence to conclude that KDM2B regulates divisions of neural progenitors? The discussion of the evidence supporting the specific effects (or roles of KDM2B) on migration, differentiation, proliferation should be improved.

(1) These are two related and important comments concerning the division and differentiation of hippocampal progenitors. We apologize for not describing and discussing clearly enough. The neurogenesis of DG neurons follows the “RGC to IPC to neurons” trajectory. At E16.5, most PAX6+ RGCs were localized at the DNe, while TBR2+ IPCs were more evenly distributed along the DNe-FDJ-DG migratory/differentiating path. Data in Fig.4 showed in E16.5 cKO hippocampi, more PAX6+ RGCs reside at the DNe and dividing more, which give rise to more TBR2+ IPCs at the DNe. In contrast, much fewer RGCs and IPCs could be detected at cKO DGs, and these RGCs and IPCs barely divide. At P7, most TBR2+ IPCs reside at the DG and divide, whereas knockout of KDM2B-CxxC results in significantly more IPCs locate at SVZ and FDJ, indicating ‘delayed neurogenesis’. In summary, the inability of RGCs and IPCs to migrate along the DNe-FDJ-DG path and to generate progeny in the DG at proper time point was summarized as “delayed differentiating trajectory”. We’ve extensively modified related texts in revision.

(2) The statement in original submission “the differentiating rate from RGCs to IPCs (PAX6+TBR2+/PAX6+) at FDJ is 60% higher in cKOs” is misleading, as only a small portion of PAX6+TBR2+ cells reside at the FDJ and DG (Fig.4o), with fewer PAX6+TBR2+ cells in cKO FDJ and DG. We’ve changed the description into “Although the differentiating rate from RGCs to IPCs (PAX6+TBR2+/PAX6+) at FDJ is 60% higher in cKOs (Fig. 4p), numbers and the ratio of dividing TBR2+ cells at FDJ decreased by 56.3% and 57.0% respectively (Fig. 4l, 4n)” in revision.

10. Please indicate in the results section what Wnt ligands were used in the

“Wnt-mix”.

We’ve indicated Wnt ligands used in the Wnt-mix in revision.

Minor comments:

- Line 139, should say Fig S2C instead of S1C.
- Line 164, should say Fig 2D-2E instead of 2C-2F.
- In the description of the Morris water maze results, it should be indicated that the probe trial was carried out on day 6.
- There is no sequential order in the description of some Figures. For example, Fig 2G is described before Fig 2D, and Fig S8 before Fig S7. This occurs in the description of several Figures and must be modified.
- Figure S3 legend, line 75 “(D)” not “(A)”.
- Fig3I, y-axis “Total” not “Totol”
- A brief description of the markers used should be given in the results section when describing the IFs (e.g. ZBTB20).

We apologize for these errors and have fixed them in revision accordingly.

Brief descriptions of the markers used have been added in revision.

Reviewer #1 (Remarks to the Author):

In this revised version of the manuscript, the authors have performed a substantial number of experiments in response to our comments. In particular, more quantifications have been added (Sup.Fig2d-p) to better describe the phenotype of mutants. Importantly, more in utero electroporations have been performed (Fig.7), strengthening the role of increased/ectopic Wnt signalling in abnormal migration and neurogenesis of DNE progenitors. We think these data have improved the manuscript and that that all our concerns have been addressed.

A minor issue that could be improved in Sup.Fig.1j would be to show split channels on the immunofluorescence so that so that Kdm2b co-localisation with TBR2 is better seen.

Reviewer #2 (Remarks to the Author):

Overall, I feel the authors have addressed our concerns, and no further questions.

Reviewer #3 (Remarks to the Author):

The authors have addressed all my comments and concerns.